# Immunotherapy with engineered bacteria by targeting the STING pathway for anti-tumor immunity

Daniel S. Leventhal[1,2✉], Anna Sokolovska [1,2✉], Ning Li[1,2], Christopher Plescia[1], Starsha A. Kolodziej[1], Carey W. Gallant[1], Rudy Christmas[1], Jian-Rong Gao[1], Michael J. James[1], Andres Abin-Fuentes[1], Munira Momin[1], Christopher Bergeron[1], Adam Fisher[1], Paul F. Miller[1], Kip A. West[1] & Jose M. Lora [1✉]

Synthetic biology is a powerful tool to create therapeutics which can be rationally designed to enable unique and combinatorial functionalities. Here we utilize non-pathogenic *E coli* Nissle as a versatile platform for the development of a living biotherapeutic for the treatment of cancer. The engineered bacterial strain, referred to as SYNB1891, targets STING-activation to phagocytic antigen-presenting cells (APCs) in the tumor and activates complementary innate immune pathways. SYNB1891 treatment results in efficacious antitumor immunity with the formation of immunological memory in murine tumor models and robust activation of human APCs. SYNB1891 is designed to meet manufacturability and regulatory requirements with built in biocontainment features which do not compromise its efficacy. This work provides a roadmap for the development of future therapeutics and demonstrates the transformative potential of synthetic biology for the treatment of human disease when drug development criteria are incorporated into the design process for a living medicine.

[1] Synlogic, Inc., 301 Binney street, suite 402, Cambridge, Massachusetts 02142, United States. [2]These authors contributed equally: Daniel S. Leventhal, Anna Sokolovska, Ning Li. ✉email: dleventhal@uchicago.edu; anna@synlogictx.com; jlora@engene.com

As the immuno-oncology field matures and as more therapies are evaluated in the clinic, our appreciation for the complexity of tumor-immune cell interactions deepens. While immunotherapies have become the standard of care for numerous cancers, significant unmet medical need persists primarily for the 55–87% of patients failing to respond to checkpoint inhibitors[1]. Combinatorial therapies seek to expand response rates[2], with an emerging interest in innate immune agonists, such as toll-like receptor (TLR)[3,4] and STimulator of INterferon Genes (STING) agonists[5–7], to drive intratumoral antigen-presenting cell (APC) activation and tumor-antigen presentation to effector T cells. Small molecule STING (smSTING) agonists are of particular interest because of their role in the production of type I interferons (IFNs) and the subsequent generation of antitumor immunity[5–7]. Recent findings are beginning to elucidate a potential shortcoming of smSTING agonists in that they may lead to activation of STING in "off-target" cell types, such as effector T cells, resulting in apoptosis[8] and ultimately impeding the formation of immunological memory[9]. Different means of targeting STING agonists to intratumoral APCs could reduce these non-targeted, systemic effects and improve the overall efficacy of such approaches.

Certain bacteria are an ideal vector for targeting STING agonism to APCs as they are actively phagocytosed and have the added benefit of triggering complementary immune pathways through the stimulation of pattern-recognition receptors (PRRs), such as the TLRs. Use of bacterial-based therapies in cancer dates back to 1891 with the identification and testing of Coley's toxin[10]. More recently, numerous studies have demonstrated the intrinsic capacity of various bacteria to selectively colonize tumors, primarily localizing to the hypoxic tumor core and sometimes leading to tumor regressions[11–17]. These therapies primarily focused on the use of attenuated pathogenic bacteria. Non-pathogenic bacteria with long historical use as probiotics and established safety profiles in humans, such as *Escherichia coli* Nissle 1917 (*Ec*N) or Symbioflor-2, represent more favorable bacterial chassis for the development of a living cancer therapeutics[17,18]. In particular, *Ec*N has a variety of advantageous features, including increased serum sensitivity[19], susceptibility to a broad range of antibiotics[18], its defined genomic landscape and most importantly its engineerability[18,20].

In this study, we evaluate the utility of *Ec*N as a platform for localized modulation of the tumor microenvironment (TME), demonstrating tumor-restricted colonization, intratumoral metabolic activity, localized immune activation, and impact on tumor growth. Then using synthetic biology techniques, we engineered a strain of *Ec*N, referred to as SYNB1891, to target STING activation to phagocytic APCs in the tumor and to trigger complementary innate immune pathways via the bacterial chassis. Synthetic biology represents a promising means to develop therapies with complex, rationally designed functionalities[21,22]. While these techniques have been utilized extensively to demonstrate proof of concept for many approaches, most fail to implement design elements which would support efficient translation into clinical evaluation. With the criteria of a clinical product at the forefront of our design process, we demonstrate the step-by-step incorporation of design elements to create a living therapeutic which produces the STING-agonist cyclic di-AMP (CDA), contains dual safety and biocontainment features and is amenable to large-scale manufacturing. Here we evaluate the mechanism of action and pharmacological properties of the final clinical candidate in relevant murine and human model systems. Importantly, SYNB1891 maintains full functionality after the incorporation of all these critical design elements which makes it safe, scalable for manufacturing and suitable for testing in humans from a regulatory perspective.

## Results

**E. coli Nissle as an oncology therapeutic vector**. In agreement with previous studies[23–27], upon intratumoral (i.t.) delivery *Ec*N expands and colonizes a wide variety of murine tumor types, including B16.F10, EL4, A20, 4T1, and CT26 syngeneic tumors in wildtype immunocompetent mice (Fig. 1a). In tumors *Ec*N exhibited a rapid expansion of 100–1000 fold, reaching steady state between 24 and 72 h, and remained localized to the tumor as bacteria were not detected in blood (Fig. 1a–c). To evaluate *Ec*N persistence and metabolic activity, we used an engineered strain containing the *LuxABCDE* bioluminescent reporter cassette (*Ec*N-Lux)[28]. Following i.t. injection, *Ec*N-Lux expanded in tumors, persisted and exhibited consistent metabolic activity (measured as bioluminescence) for up to 14 days (Fig. 1b). Bioluminescence also enabled qualitative monitoring of bacterial localization within the tumor and surrounding tissue on a macroscopic scale. Following i.t. injection bioluminescence is detected within and throughout the tumor mass; however, it is not detected in the subcutaneous space surrounding the tumor up to 72 h post-injection (Supplementary Fig. 1a). Intratumoral treatment resulted in dose-dependent increases in IL-6 and TNFα in both tumor and serum at early time points (Fig. 1d, e and Supplementary Fig. 1b, c), yet, the magnitude and duration of the response were substantially higher within tumors. Finally, i.t. administration of *Ec*N resulted in a significant delay in tumor growth compared to saline injection controls but did not result in complete tumor regressions (Fig. 1f). Collectively, these data demonstrate the utility of *Ec*N as a delivery vector for the treatment of cancer which exhibits localization, persistence, metabolic activity within tumor tissue and a moderate level of anti-tumor activity. Such attributes would be important for the sustained delivery of relevant immunological therapeutic payloads.

**Engineering a STING agonist-producing live therapeutic**. STING bridges innate and adaptive immunity through activation of APCs, production of type I IFNs, neoantigen cross-presentation to cytotoxic T cells and subsequently the initiation of tumor-specific T cell responses[5–7]. Cyclic di-nucleotides (CDNs) bind to and activate STING, triggering type I IFNs[29,30]. Physiologically, CDNs can originate from host cyclic GMP-AMP synthase (cGAS) following detection of cytosolic double-stranded DNA[29] or when produced by invading, intracellular bacteria. While the CDN product of cGAS, 2′3′-cGAMP, is made by eukaryotic cells, prokaryotes produce and utilize other CDNs, such as CDA, cyclic-di-GMP and 3′3′-cGAMP, for critical signaling processes[31]. We developed a STING-agonist-producing *Ec*N therapeutic strain by systematically prototyping and incorporating the various components necessary for a therapy intended for large-scale manufacturing and translation to the clinic. These steps included selection of a CDN-producing enzyme, selection of an inducible promoter, incorporation of auxotrophies for safety and biocontainment, and removal of any antibiotic resistances used during strain engineering and development.

A panel of three CDA-producing enzymes (encoded by *dncV* from *Vibrio cholerae*, *cdaS* from *Bacillus subtilis* and *dacA* from *Listeria monocytogenes*) were expressed in *Ec*N under a tetracycline-inducible promoter ($P_{tet}$) and the resulting levels of CDA production were evaluated in vitro. Diadenylate cyclase (DacA) was selected for further evaluation as it produced the highest levels of CDA following gene induction in *Ec*N (Fig. 2a). Co-culture of *dacA*-expressing *Ec*N (SYN-$P_{tet}$-*dacA*) with the RAW 264.7 immortalized macrophage cell line resulted in a dose-dependent secretion of IFNβ1, whereas non-induced SYN-$P_{tet}$-*dacA* did not (Fig. 2b). To evaluate activity in vivo, SYN-$P_{tet}$-*dacA* or non-engineered *Ec*N were i.t. administered followed by

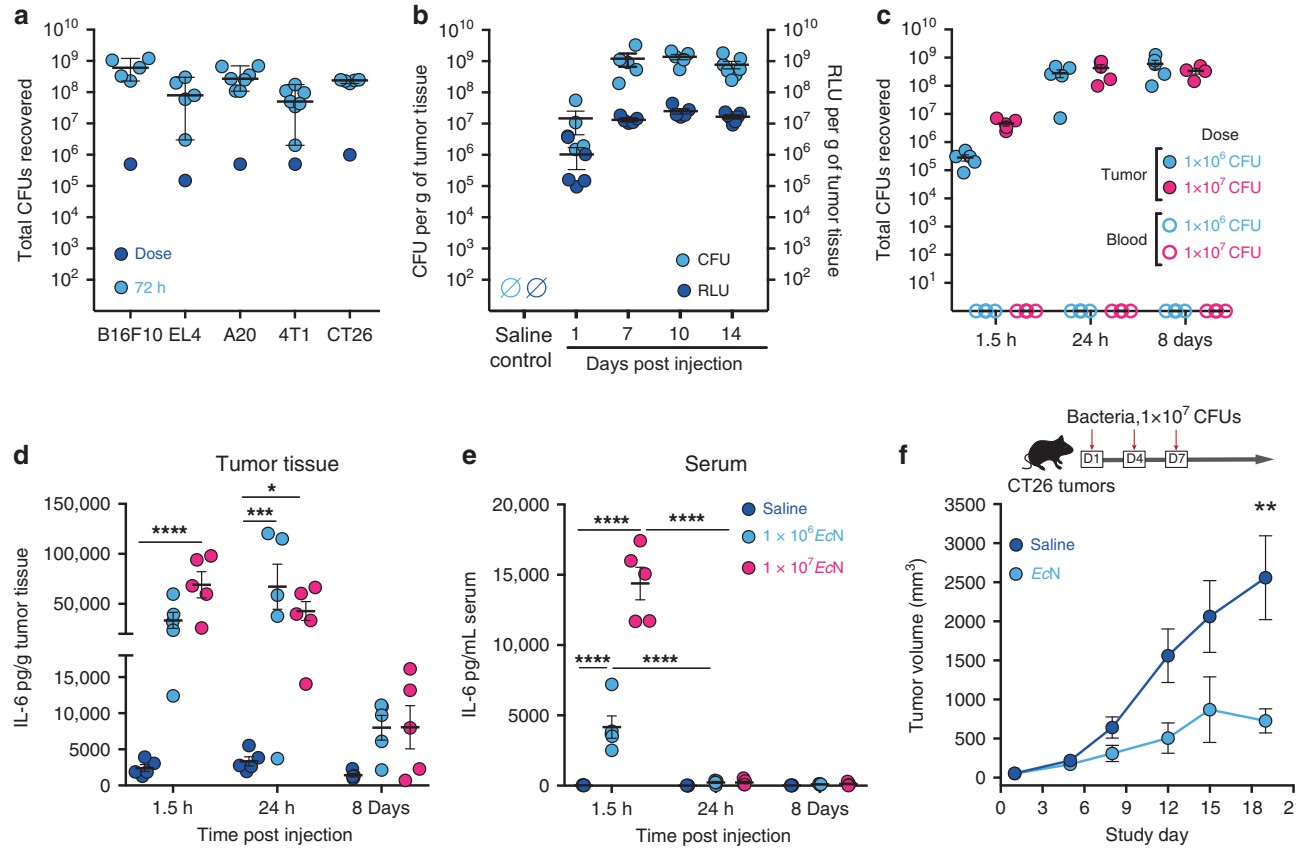

**Fig. 1 *E coli*. Nissle is a versatile platform for localized modulation of the tumor microenvironment. a** Initial dose and bacterial abundance within tumor homogenates at 72 h post-intratumoral (i.t.) injection with ~1 × 10⁶ CFU of *Ec*N from the indicated tumor model as measured by colony forming unit (CFU) assay (B16F10, EL4, CT26 *n* = 5 mice per group, A20, 4T1 *n* = 7 mice per group). Data are representative of two independent experiments. **b** Bacterial abundance measured by CFU (left axis) and relative bioluminescent units (RLU) (right axis) from CT26 tumors at the indicated time points post-i.t. injection of *n* = 5 mice with 1 × 10⁶ CFU of *Ec*N-*LuxABCDE* compared to saline injected controls. Data are representative of two independent experiments. **c–e** CT26 tumor-bearing mice (*n* = 5 per group) were treated i.t. once with indicated doses of bacteria. Bacterial abundance within tumor homogenates (filled in circle) or blood (hollow circle) for mice treated with 1 × 10⁶ or 1 × 10⁷ CFUs *Ec*N (**c**), and IL-6 quantification from tumor (**d**) and serum (**e**) shown at the indicated time points (**P* = 0.0175, ****P* = 0.0001, *****P* < 0.0001, two-way ANOVA with Tukey's multiple comparisons tests). Data are representative of two independent experiments. **f** Representative tumor growth data for CT26 tumor-bearing mice i.t. treated with either saline control or 1 × 10⁷ CFUs of *Ec*N (*n* = 9 mice per saline and *n* = 7 mice per *Ec*N group) on days 1, 4 and 7 (***P* = 0.0062, two-tailed unpaired Student's *t* test comparing saline vs *Ec*N treated groups on day 19 of study). Individual tumor volumes are presented in Supplementary Fig. 1d. Data are representative of three independent experiments Each circle in (**a–e**) represents an individual animal. **a–f** Data are mean with s.e.m.

intraperitoneal (i.p.) injection of anhydrous tetracycline (aTc) into B16.F10 melanoma tumor-bearing mice. B16.F10 was selected as a relevant tumor models as this represents a poorly inflamed tumor which is typically refractory to immunotherapy in mice[32,33]. Similar to results obtained in vitro, production of CDA by engineered *Ec*N resulted in a gain of IFNβ1 induction, compared to non-engineered *Ec*N, in vivo (Fig. 2c). Additionally, SYN-P_tet-*dacA* treatment significantly decreased tumor growth by eight days post-treatment initiation (Supplementary Fig. 2a–c). While treatment with both engineered and non-engineered *Ec*N led to early increases in innate associated cytokines (like TNFα, GM-CSF, IL-6, IL-1β and CCL2) 24 h post dose initiation (Supplementary Fig. 2d), only SYN-P_tet-*dacA* treatment resulted in a shift to expression of T-cell associated cytokines (like IL-2, Granzyme B and IFNγ) at 8 days post dose initiation (Supplementary Fig. 2e). Collectively, these data suggest the engineered expression of CDA in *Ec*N enables the additional activation of type I IFNs and possibly the initiation of efficacious, T-cell antitumor immunity in vivo.

Tightly controlled induction of the *dacA* gene circuit is critical from a manufacturing perspective as rapid depletion of adenosine triphosphate (ATP) and/or production of the bacterial signaling molecule CDA could hinder large-scale biomass production and bacterial fitness, and as such the use of an inducible promoter becomes crucial. Since the utilization of tetracyclines as an induction agent is not desirable for clinical studies, we evaluated several other promoter/inducer systems. To assess activity in vivo we introduced inducible promoter-GFP cassettes into an *Ec*N strain which constitutively expressed mCherry (RFP) and systemically administered induction agents following bacterial colonization of B16.F10 tumors. This approach allowed for the detection of RFP⁺ bacteria by flow cytometry from tumor cell suspensions and quantification of gene induction by GFP (Fig. 2d). Utilizing this approach, we evaluated the activity of the salicylate (P_sal), cumate (P_cmt) and fumarate-and-nitrate reductase (P_fnrS) promoters, in combination with their associated agonists, salicylic acid, cumate, and hypoxia, respectively. While all promoters showed functionality within an hour of administration of their respective inducing agent, the hypoxia inducible P_fnrS[34,35] was selected for further development as this promoter showed the highest levels of intratumoral payload induction, with 65–95% of all bacteria induced. These data are in agreement with

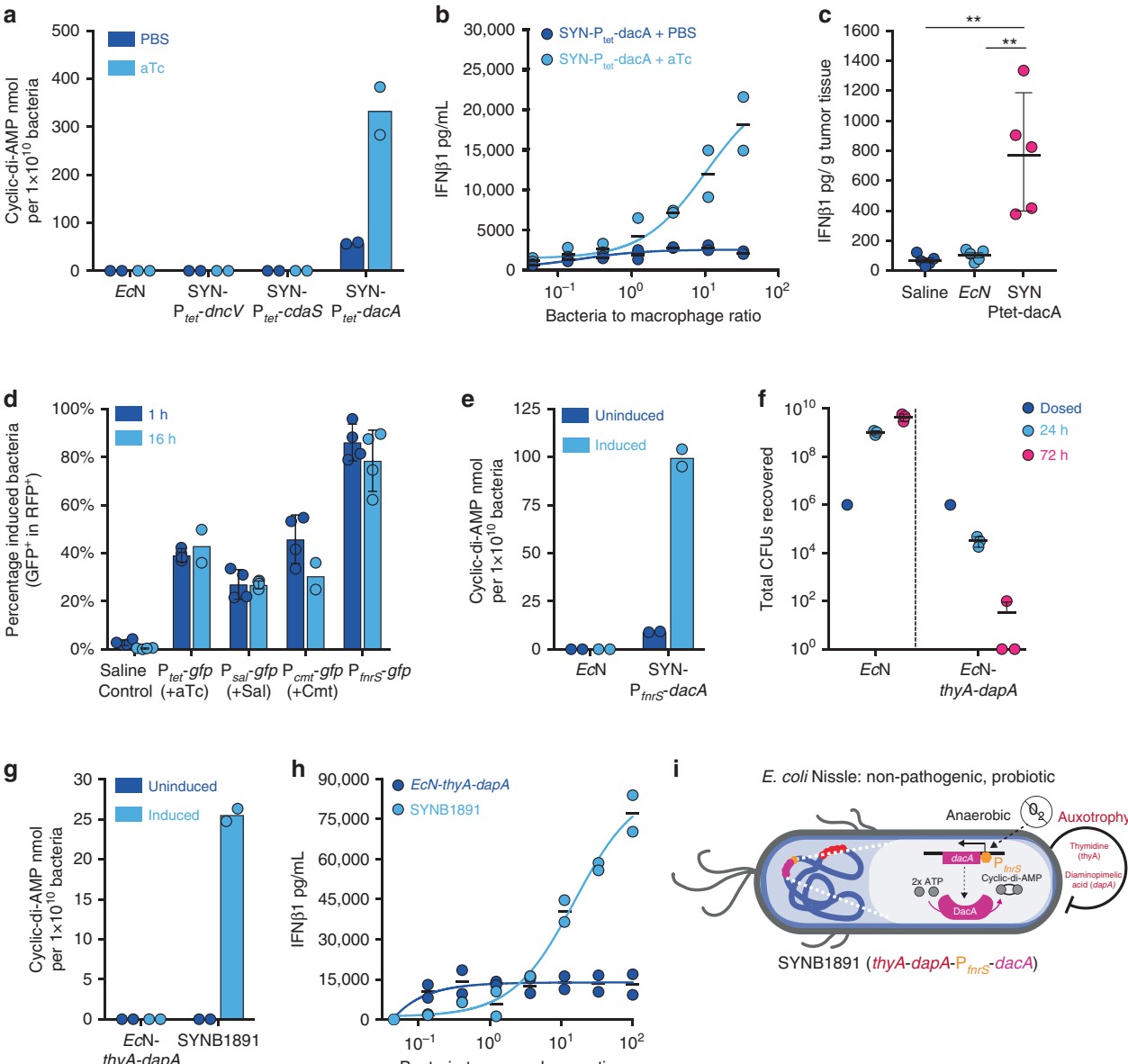

**Fig. 2 Engineering a STING agonist-producing live therapeutic (SYNB1891).** **a** Cyclic di-AMP (CDA) abundance from bacterial cell pellets of the indicated strains cultured with PBS (non-induced) or 100 ng/mL aTc (induced) for 4 h. Data are representative of three independent experiments. **b** IFNβ1 production by RAW 264.7 cells co-cultured for 4 h with non-induced (+PBS) or pre-induced (+aTc) SYN-$P_{tet}$-dacA. **c** B16.F10 tumor-bearing mice were i.t.-injected with $1 \times 10^6$ CFUs of EcN or SYN-pTet-dacA bacteria, or saline control ($n = 5$ mice per group). Four hours later all mice received 10 μg aTc intraperitoneally. Intratumoral IFNβ1 is shown at 24 h post-injection (**$P = 0.0011$ (vs saline), **$P = 0.0016$ (vs EcN), one-way ANOVA with Tukey's multiple comparisons tests). Additional data are shown in Supplementary Fig. 2a–e. **d** B16.F10 tumor-bearing mice were i.t.-injected with saline or $5 \times 10^6$ CFUs of bacterial strains containing a constitutively expressed mCherry (RFP) and inducible GFP ($n = 4$ mice per group except $n = 2$ for groups 16 h $P_{tet}$-gfp and 16 h $P_{cmt}$-gfp). To induce GFP mice were injected with either saline (control), aTc, sodium salicylate (Sal) or p-isopropylbenzoate (Cmt). Percentage of induced bacteria (GFP+ among RFP+ cells) in tumors is shown at indicated times post-induction. **e** CDA abundance from bacterial cell pellets of the indicated strains cultured under aerobic (uninduced) or anaerobic (induced) conditions. **f** Total CFUs recovered from B16.F10 tumors i.t.-injected with $1 \times 10^6$ CFUs of either prototrophic EcN or a strain containing dual thyA and dapA auxotrophies at the indicated time-points ($n = 3$ mice per group per time point). **g** CDA abundance from bacterial cell pellets of the indicated strains cultured under aerobic (uninduced) or anaerobic (induced) conditions. Data are representative of five independent experiments. **h** IFNβ1 production from 4-h macrophage and bacterial cell co-cultures, as described in (**b**), with the indicated bacterial strains. **i** Schematic of the finalized SYNB1891 bacteria strain containing all engineering components. **a**, **b**, **g**, **h** $n = 2$ biological replicates per group per time point. **c**, **d**, **f** Mean and s.d. shown. **b**–**f**, **h** Data are representative of two independent experiments. Each circle represents an individual animal or independent experimental replicate.

the results reported for *Salmonella typhimurium* where a set of hypoxia sensitive promoters, such as the Nitrate Reductase promoter, were shown to enable tumor-specific induction compared to the spleen[36]. Furthermore, $P_{fnrS}$ does not require administration of an exogenous agent due to the hypoxic nature of the TME and oxygen concentration can be tightly controlled during bacterial expansion in fermenters. The feasibility of hypoxia inducible CDA production was confirmed for EcN containing the $P_{fnrS}$-dacA circuit under anaerobic conditions in vitro (Fig. 2e).

From a safety and regulatory perspective, biocontainment controls are critical elements of a bacterial-based live therapeutic for clinical use[37,38]. The introduction of a thymidine (thy) auxotrophy by deletion of the thymidylate synthase gene (*thyA*) has been shown to be effective as a biocontainment mechanism, as free thy is not readily available in most extracellular spaces[37]. We found that the TME contains a sufficient concentration of free thy for the *Ec*N *thyA* mutant to proliferate and colonize (Supplementary Fig. 2f). Diaminopimelic acid (dap) is a component of the bacterial cell wall and is not produced by eukaryotes, and therefore we hypothesized that a dap auxotrophic strain would be unable to survive in a mammalian host environment. Indeed, deletion of the *dapA* gene (encoding 4-hydroxy-tetrahydropicolinate synthase) resulted in a mutant strain of *Ec*N that was unable to expand or persist within tumors and was cleared over time (Supplementary Fig. 2f). We thus engineered a dual safety mechanism by introducing both *dapA* and *thyA* deletions to prevent intratumoral and extra-tumoral bacterial proliferation, respectively, and the inability of a double mutant to proliferate in vivo was confirmed in a variety of tumor types (Fig. 2f and Supplementary Fig. 2g).

To ensure stability during manufacturing and to meet regulatory guidelines the P$_{fnrS}$-*dacA* circuit was inserted into the genome of the *dapA*, *thyA* double mutant and all antibiotic resistance genes were removed. The final clinical candidate strain, referred to as SYNB1891, maintained its anaerobically inducible production of CDA (Fig. 2g), dose-dependent biological activity when co-cultured with RAW 264.7 macrophage cells in vitro (Fig. 2h), inherent sensitivity to human serum (Supplementary Fig. 2h) and sensitivity to a wide panel of antibiotics currently utilized in the clinic (Supplementary Fig. 2i). In summary, the selection and validation of these various modular components constituted our clinical candidate strain, SYNB1891, consisting of an engineered *Ec*N with dual *dapA* and *thyA* auxotrophies, the genomic integration of the *dacA* gene under the control of the anaerobically inducible P$_{fnrS}$ promoter and removal of antibiotic resistance genes (Fig. 2i).

**Mechanisms of SYNB1891-mediated type I interferon induction.** To begin deconvoluting the mechanisms of action of SYNB1891, we treated murine bone marrow-derived dendritic cells (BMDCs) from wild-type (WT), *Tmem173*$^{gt}$ (STING$^{-/-}$) or *Tlr4*$^{lps-del}$ (TLR4$^{-/-}$) mice and incorporated relevant pathway controls, such as purified lipopolysaccharide (LPS) and a benchmark selective smSTING-agonist, 2′3′-c-di-AM(PS)2 (Rp, Rp), utilized at a concentration (5 μg/ml) similar to what has been previously reported[6]. Type I interferon production by BMDCs in response to SYNB1891 was greatly dependent on STING signaling, as STING$^{-/-}$ BMDCs failed to induce high levels of IFNβ1 expression (Fig. 3a and Supplementary Fig. 3a). *Ec*N chassis itself and LPS induced low levels of IFNβ1 expression in a TLR4-dependent but STING-independent manner (Fig. 3a and Supplementary Fig. 3a), suggesting that TLR4 activation may play a minor role in type I IFN induction in response to SYNB1891. IFNβ1 expression in response to smSTING agonist was preserved in TLR4$^{-/-}$ BMDCs and lost in STING$^{-/-}$ BMDCs (Fig. 3a and Supplementary Fig. 3a). Additionally, SYNB1891 and smSTING-agonist treatment resulted in similar levels of upregulation for the co-stimulatory marker *cd86* as compared to controls (Supplementary Fig. 3b). Despite a peak ~1.5-fold increase in IFNβ1 production compared to SYNB1891 (Fig. 3a), smSTING-agonist resulted in far less or no production of additional proinflammatory cytokines like IL-6, IL-12, TNFα and IL-1β up to 18 h post-treatment (Fig. 3b, c and Supplementary Fig. 3c–g). LPS-TLR4 signaling may contribute to the expression of these additional

cytokines, as lack of TLR4 signaling significantly blunted expression of *Il6* and *Il12a* mRNA in response to SYNB1891, control *Ec*N and LPS (Fig. 3b, c). Collectively, SYNB1891 treatment resulted in the greatest production for all analyzed cytokines relative to *Ec*N control, suggesting that both the bacterial chassis and the engineered expression of CDA play important roles in the overall immune-activating mechanism of action of SYNB1891 (Supplementary Fig. 3c–g).

Next, we used the phagocytosis inhibitor cytochalasin D to evaluate the contribution of active cellular uptake of SYNB1891 towards its overall mechanism of action in macrophages (RAW cells) and BMDCs. Cytochalasin D inhibits actin polymerization and prevents phagocytosis, yet it has minimal effect on endocytosis or pinocytosis of soluble small molecules[39,40]. To quantify phagocytosed bacteria, nuclei and F-actin staining were used to identity individual BMDCs and demarcate their cellular border, respectively. Co-culture of murine BMDCs with SYNB1891 modified to express GFP (SYNB1891-*gfp*), showed many bacterial cells were internalized (e.g. co-localized with F-actin staining) and residing within mature phagosomes that contained lysosome-associated membrane protein (LAMP-1) (Fig. 3d, e). Pretreatment with cytochalasin D significantly inhibited the number of bacterial cells observed within BMDCs (Fig. 3f, g) and abrogated expression of IFNβ1 following treatment with SYNB1891 in both cell types (Fig. 3h, j and Supplementary Fig. 3h, k). In contrast, cytochalasin D did not significantly impact IFNβ1 induction following treatment with smSTING-agonist (Fig. 3h, j and Supplementary Fig. 3h, k). Inhibition of phagocytosis did not significantly impact TNFα production in response to bacterial or LPS treatment (Supplementary Fig. 2j), yet decreased IL-6 expression by 2.5-5 fold (Fig. 3i, k). These data suggest that *Ec*N can trigger proinflammatory cytokines secretion from the surface of the phagocytic cell, although internalization is required to optimally enhance signaling, induce IFNβ1 and activate STING. Relative expression levels of each cytokine and the impact of phagocytosis on their expression by *Ec*N, SYNB1891 and smSTING-agonist treatments are summarized in Supplementary Fig. 3l.

Collectively, these data suggest that SYNB1891 is phagocytosed and active uptake is required for STING-dependent induction of type I IFN responses, thus providing a natural mechanism for preferential activation of APCs. Moreover, the data demonstrate the pleiotropic activity provided by the *Ec*N chassis of SYNB1891 which activates parallel innate immune signaling pathways, through PRRs like TLR4, and results in the expression of complementary proinflammatory cytokines.

**SYNB1891 triggers multiple innate immune pathways in vivo.** To evaluate its activity in vivo, we delivered a single i.t. injection of SYNB1891 at various dose levels into established B16.F10 murine melanoma tumors and evaluated its pharmacokinetics and pharmacodynamics over time. As expected, the *dapA* mutation incorporated into SYNB1891 resulted in its clearance from the TME over time, with bacteria undetectable by colony forming unit (CFU) assay in blood at all time points (Fig. 4a and Supplementary Fig. 4b, c). Quantitative PCR (qPCR) using SYNB1891 specific probes was utilized as a secondary means to detect bacteria presence in the blood independent of cell viability. While SYNB1891 DNA was detected in the majority of mice injected intravenously (as a positive control) up to 24-h post dose, only 1 of 5 mice receiving i.t. injection at the highest dose had detectable levels of SYNB1891 in blood at 2- and 6-h (Supplementary Fig. 4d, e) post-injection. SYNB1891 treatment delayed tumor growth and resulted in undetectable tumors in some animals (Fig. 4b). Intratumoral production of CDA was detected at

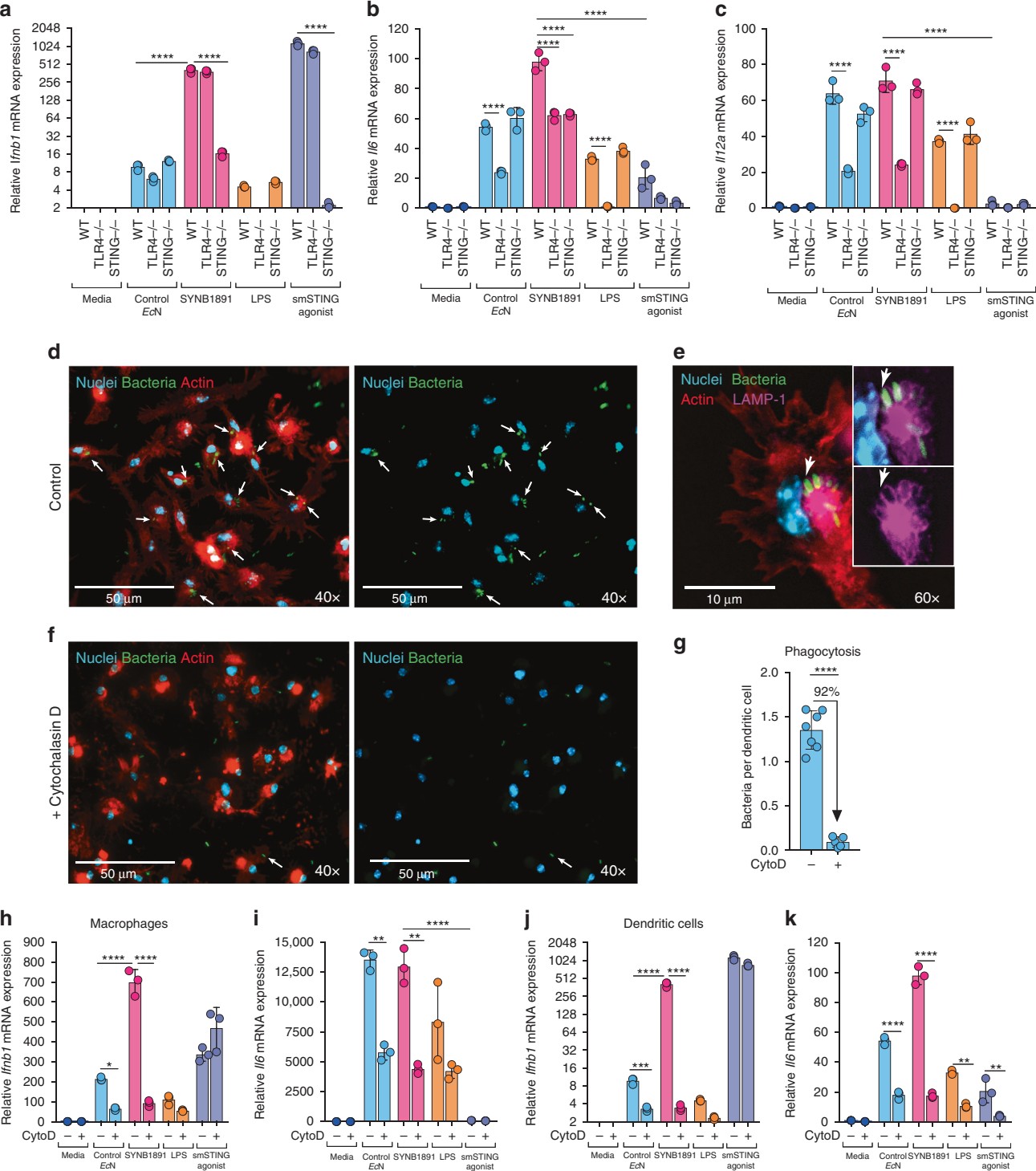

24- and 72-h post dose, confirming the functional activity of SYNB1891 in vivo (Fig. 4c). Type I IFNs (Fig. 4d, e) as well as a variety of proinflammatory cytokines, including TNFα, IL-6, IL-1β, IFNγ and GM-CSF (Fig. 4f–j), were induced in a dose-dependent manner, highlighting the dual activity of both CDA and the EcN bacterial chassis. In agreement with previous reports utilizing smSTING-agonists[41], significant upregulation of IL-15 was also observed following treatment with SYNB1891 (Fig. 4k). Collectively, these data demonstrate the dose-dependent pharmacology of SYNB1891, its target engagement in vivo and its utility as a potent inducer of localized inflammation and its potential anti-tumor activity.

**SYNB1891 generates efficacious anti-tumor immunity.** Durable responses are a hallmark of successful immunotherapy. To evaluate the efficacy of SYNB1891 we treated B16.F10 tumor-bearing mice with three i.t. injections over the course of a week and monitored them long-term. SYNB1891 treatment resulted in a dose-dependent significant delay in tumor growth compared to saline injection controls and drove durable tumor rejections for 30–40% of animals (Fig. 5a and Supplementary Fig. 5a–c). While injection of control EcN resulted in a significant delay of tumor growth, it did not result in complete tumor rejection. Utilizing a benchmark smSTING-agonist at a previously reported efficacious dose (50 μg per animal)[6], we performed a head-to-head

**Fig. 3 Phagocytosis- and STING-dependent induction of type I interferon by SYNB1891. a–c** WT, TLR4$^{-/-}$ and STING$^{-/-}$ BMDCs were treated with Control EcN (MOI: 25), pre-induced SYN1891 (MOI: 25), LPS (100 ng/mL), or smSTING agonist (5 μg/mL) for 4 h ($n = 3$ biological replicates per group per genotype). BMDCs from each genotype incubated alone served as negative controls. Cells were analyzed for the upregulation of *Ifnb1* (**a**), *Il6* (**b**) and *Il12a* (**c**) mRNA (****$P < 0.0001$ two-way ANOVA with Tukey's multiple comparisons tests, see Supplementary Fig. 3a for IFN-β1 protein quantification). **d–g** Representative fluorescent images and phagocytosis quantification. WT BMDCs were incubated with pre-induced SYNB1891-*gfp* (MOI: 25) for 1 h in control media (**d**, **e**) or pre-treated for 1 h with Cytochalasin D (10 μM) before bacterial incubation (**f**). Non-internalized bacteria were washed out and cells were stained for microscopy. Cell nuclei were labeled with Hoechst (Blue), F-actin was stained with ActinRed 555 probe (Red), SYNB1891-*gfp* were labeled with anti-GFP (Green), and phagosomal transmembrane protein LAMP-1 was labeled with anti-LAMP1 (Purple). White arrows point to bacteria co-localized within DC actin cytoskeleton (**d–f**). These bacteria are surrounded by phagosomal membrane stained with LAMP-1 (**e**). The number of bacteria per dendritic cell per field of view (FOV) was quantified in (**g**). 12 FOVs were evaluated in the experiment (7 for control and 5 for CytoD group) resulting in >200 cells per treatment quantified (**$P < 0.0001$, two-tailed unpaired Student's *t* test). Images and data are representative of 2 independent experiments. **h**, **i** RAW 264.7 macrophages or (**j**, **k**) WT BMDCs were treated as described above in (**a**) ($n = 3$ biological replicates per group). In the indicated groups cells were pre-treated for 1 h with Cytochalasin D (10 μM). Macrophages or BMDCs incubated in media alone served as a negative control. Cells were analyzed for the upregulation of *Ifnb1* (**g**, **i**) and *Il6* (**h**, **j**) mRNA (*$P = 0.0182$, **$P = 0.0017$-0.006, ***$P = 0.0005$, ****$P < 0.0001$ two-way ANOVA with Tukey's multiple comparisons tests, see Supplementary Fig. 3h–k for protein quantification,). **a–c**, **g–k** Data are representative of two or more independent experiments per cell type with mean and s.d. shown. Each circle represents an independent experimental replicate.

comparison of the long-term efficacy of the bacterial chassis alone (Control EcN), STING activation alone (smSTING Agonist) or engineered STING agonist in a bacterial chassis (SYNB1891). While both STING-agonist approaches were effective at controlling tumor growth for up to 2 weeks, SYNB1891 treatment resulted in greater long-term efficacy (40% survival) compared to treatment with smSTING agonist (10% survival) (Fig. 5b and Supplementary Fig. 5d, e).

To extend our findings to additional tumor types, we performed similar studies on A20 B cell lymphoma tumors. Treatment of A20 tumors with SYNB1891 resulted in a dose-dependent effect on tumor control with $10^8$, $5 \times 10^8$ and $10^9$ CFU doses resulting in 30%, 50%, and 80% of mice with complete tumor rejections, respectively (Fig. 5c and Supplementary Fig. 6a, b). To evaluate the contribution of T cells towards SYNB1891 efficacy in the A20 model, we depleted CD4$^+$ T cells or CD8$^+$ T cells prior-to treatment initiation and throughout the course of the study using depleting antibodies. While mice treated with either isotype control or a CD4$^+$ T cell-depleting antibody exhibited a 40-50% complete response rate, 0% of mice receiving a CD8$^+$ T cell-depleting antibody survived long-term (Fig. 5d). Finally, SYNB1891-treated animals that had remained A20 tumor-free for at least 60 days were rechallenged with A20 cells on the contralateral flank. Unlike age-matched naïve controls which showed rapid tumor growth, all cured mice remained tumor-free after re-challenge (Fig. 5e and Supplementary Fig. 6c). These data suggest that SYNB1891 treatment results in the activation of CD8$^+$ T cells which are required for long-term efficacy and likely the formation of protective immunological memory.

Collectively, these data demonstrate the complementary contributions of both the EcN bacterial chassis and the production of the STING agonist CDA towards the generation of an efficacious and durable antitumor immune response in two distinct murine tumor models (B16F10 and A20) in two genetic backgrounds (C57BL/6 and BALB/c).

**SYNB1891 activates multiple human STING alleles.** To assess activity of SYNB1891 in human APCs, we first evaluated STING pathway induction utilizing modified human monocyte THP-1 cell lines containing an interferon-regulatory factor (IRF) reporter. SYNB1891 treatment resulted in robust dose-dependent IRF induction which was lost in STING$^{-/-}$ cells (Fig. 6a) and reduced in cGAS$^{-/-}$ cells (Fig. 6b). In contrast to murine APCs where bacterial chassis did not induce significant amount of IFN-β1, treatment of human THP-1 cells with non-engineered control EcN led to moderate, dose-dependent STING pathway induction

(Fig. 6b). However, bacterial chassis alone completely lost the ability to stimulate IRF in cGAS$^{-/-}$ cells. (Fig. 6b). These data suggest a dual mechanism of action for SYNB1891 in the stimulation of STING in human APCs which includes direct activation of STING via CDA produced by SYNB1891, and activation of cGAS, likely through detection of cytoplasmic bacterial DNA (Supplementary Fig. 7a). Treatment of THP-1 cells containing an NF-κB reporter with SYNB1891 also led to a robust dose-dependent activation of the NF-κB pathway, which was partially reduced, ~1.6-fold, in STING$^{-/-}$ cells (Fig. 6c). These results also suggest that SYNB1891 triggers NF-κB induction through STING-dependent and STING-independent pathways.

Next, we evaluated STING-pathway activity across a panel of THP-1 cell lines which contained three of the most prevalent human *TMEM173* (STING) variants: R232 representing 57.9%, HAQ representing 20.4% and H232 representing 13.7% of alleles found in the human population, respectively[42]. SYNB1891 treatment resulted in ~9-to-28-fold IRF induction across all three alleles compared to non-stimulated or STING$^{-/-}$ controls (Fig. 6d). In agreement with previous studies[6], the HAQ allele showed the highest level of induction with the R232 and H232 alleles showing intermediate and lower activity, respectively. Next, we evaluated SYN1891 activity and the contribution of phagocytosis in primary human, monocyte-derived dendritic cells (DCs). Similar to results obtained in murine APCs, treatment of primary human DCs with SYNB1891 resulted in robust expression of *IFNB1*, the expression of various proinflammatory cytokines (such as *IL6, TNFa, IL-1B,* and *IL12A*) and the upregulation of various DC maturation makers (like *CD86* and *CD40*) (Fig. 6e, f and Supplementary Fig. 7b–l). While treatment with either control EcN or SYNB1891 resulted in similar levels of expression for the various proinflammatory cytokines and DC maturation markers, SYNB1891 treatment led to significantly higher induction of *IFNB1*. Both *IFNB1* and *IL6* expression were decreased following pretreatment with cytochalasin D, suggesting that optimal stimulation of the STING and NF-κB pathways requires active bacterial phagocytosis by human APCs (Fig. 6e, f). Treatment with up to 25 μg/mL smSTING-agonist resulted in comparable levels of *CD86* expression, however lower levels of *IFNB1, CD40, IL6,* and *IL12A* expression relative to SYNB1891, with no detection of *IL1B* above background (Fig. 6e, f and Supplementary Fig. 7b–l).

Together, these results confirm the translation of our findings from mouse models to human APCs for the induction of type I IFNs and NF-κB-associated cytokines following SYNB1891 treatment, and for a range of *TMEM173* allelic variants representing a large majority of the human population.

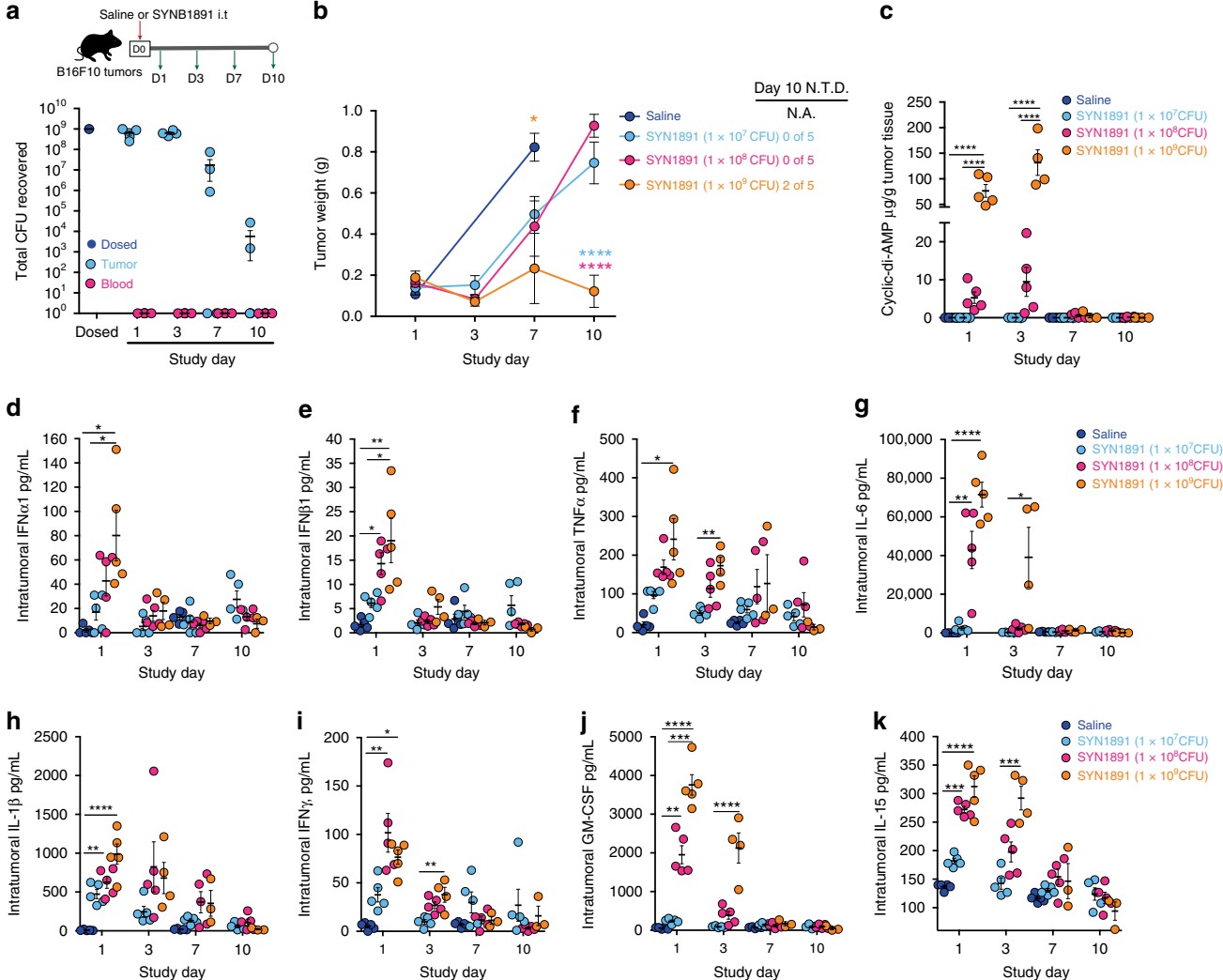

**Fig. 4 In vivo dynamics of SYNB1891 in tumor bearing mice. a–k** B16.F10 tumor-bearing mice were treated with a single i.t. dose of either saline, $1 \times 10^7$, $1 \times 10^8$ or $1 \times 10^9$ CFUs SYNB1891 on Study Day 0 ($n = 5$ injected mice per bacterial group per time point, for Saline group $n = 5$ injected mice for day 1 and day 7 analyzes, for the bacterial group $1 \times 10^9$ CFUs SYNB1891 for day 3 $n = 4$ analyzed tumors, for days 7 and 10 $n = 3$ analyzed tumors due to no visible tumor mass upon dissection). **a** Bacterial abundance within tumor homogenates and blood at the indicated timepoints post-i.t. injection with $1 \times 10^9$ CFUs (see Supplementary Fig. 4b, c for $1 \times 10^7$ and $1 \times 10^8$ doses). **b** Total tumor weight for treated mice at the indicated time points with mice having no visible tumor mass upon dissection (no tumor detected (N.T.D)) on Study day 10 (* $P = 0.0163$ (orange stars—indicated group vs $1 \times 10^9$ CFUs SYNB1891), one-way ANOVA with Tukey's multiple comparisons tests for Day 7; ****$P < 0.0001$ (blue stars—indicated group vs $1 \times 10^8$ CFUs SYNB1891, pink stars—indicated group vs $1 \times 10^7$ CFUs SYNB1891) two-way ANOVA for bacterial groups with Tukey's multiple comparisons tests). Individual tumor volumes are presented in Supplementary Fig. 4a). **c** Cyclic-di-AMP abundance from tumor homogenates, and, (**d–k**), cytokine abundance from tumor supernatants for IFNα1 (**d**), IFNβ1 (**e**), TNFα (**f**), IL-6 (**g**), IL-1β (**h**), IFNγ (**i**), GM-CSF (**j**) and IL-15 (**k**) from treated mice at the indicated time points post-i.t injection (*$P = [0.016–0.031]$, **$P = [0.0011–0.0041]$, ***$P = [0.0001–0.0003]$, ****$P < 0.0001$ one-way ANOVA with Tukey's multiple comparisons tests at indicated time points). Data are representative of two independent experiments with mean and s.e.m. shown. Each circle in (**a**) and (**c–k**) represents an individual animal.

## Discussion

In this study, we used synthetic biology techniques and drug development criteria to design and create an engineered bacterial strain capable of localized, targeted STING activation and a product suitable for manufacturing and evaluation in human clinical trials. SYNB1891's functionality was rationally designed by incorporating the *dacA*, CDA-producing enzyme, in a tumor tropic bacterial chassis to target STING activation to APCs in the TME. In addition, as a Gram-negative bacterial chassis *Ec*N triggers complementary proinflammatory pathways through PRR activation. Safety and biocontainment features were incorporated by controlling bacterial proliferation inside and outside of tumors through two distinct auxotrophies, *thyA* and *dapA*, and by

removing antibiotic resistance genes to enable clearance of bacteria by available treatments. Lastly, by utilizing the $P_{fnrS}$ promoter to control *dacA* expression, STING-agonist production is repressed to generate biomass during manufacturing and later induced upon entry into the hypoxic TME.

The resulting biotherapeutic, SYNB1891, was shown to produce high levels of CDA and induce potent type I IFN production in a phagocytosis-dependent manner in both, mouse and human APCs. Mechanistically, smSTING agonists non-specifically penetrate a variety of cell types while a bacterial vector like SYNB1891 preferentially reaches the intracellular space of APCs via active phagocytosis. This is potentially an important feature to avoid "off-target" STING-pathway activation in non-APCs, such

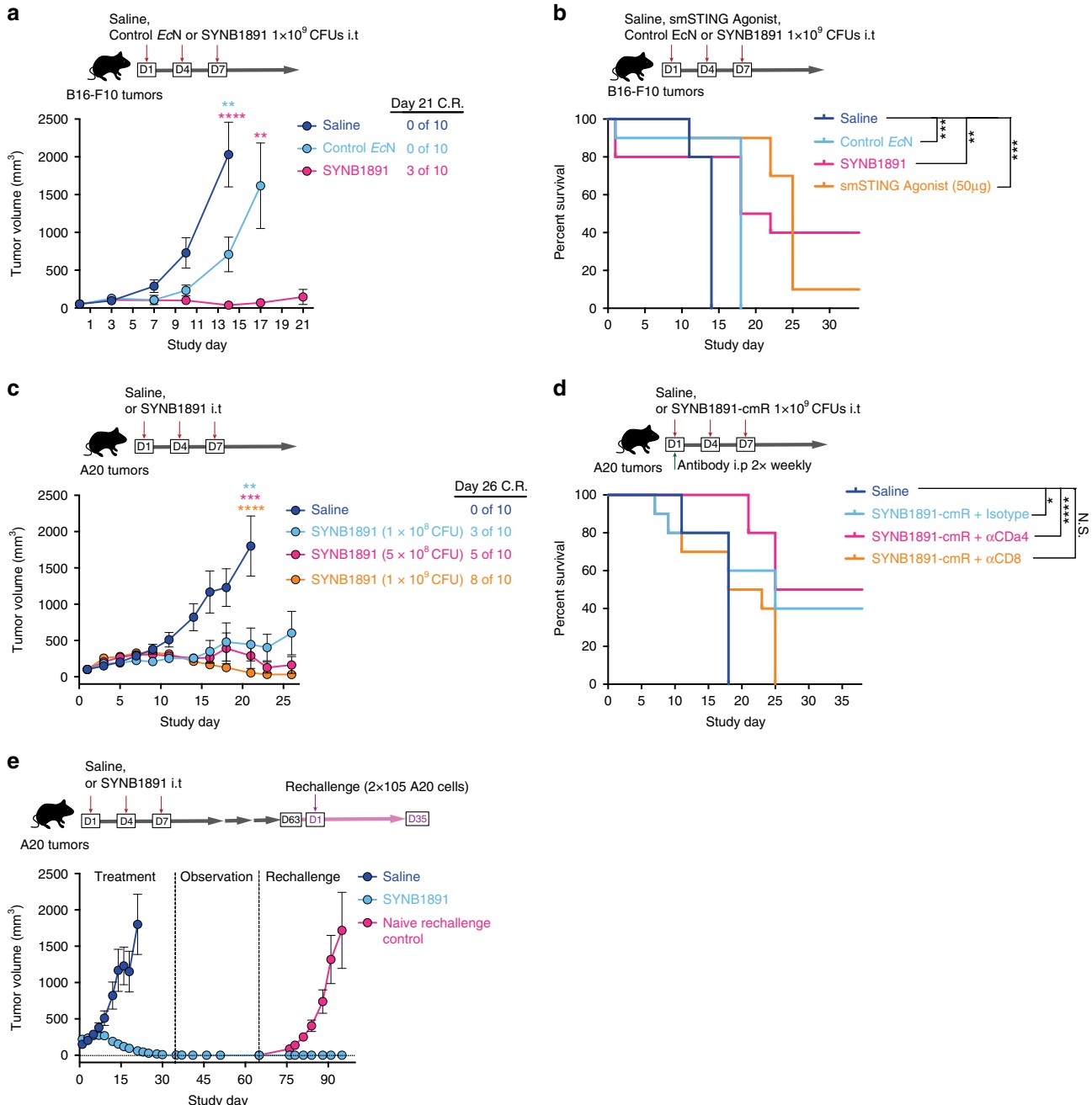

**Fig. 5 SYNB1891 treatment triggers efficacious antitumor immunity and immunological memory. a** On Study days 1, 4, and 7, B16.F10 tumor-bearing mice were treated i.t. with either saline (control), SYNB1891 or Control *Ec*N lacking the P$_{fnrs}$-*dacA* circuit. Tumor growth data is shown with the ratio of complete responders (C.R.). (**$P = 0.0058$ (blue stars—indicated group vs *Ec*N), ****$P < 0.0001$ (pink stars—indicated group vs SYNB1891) one-way ANOVA with Tukey's multiple comparisons tests for Day 14; **$P = 0.0078$ (pink stars—*Ec*N vs SYNB1891, two-tailed unpaired Student's *t* test. Individual tumor volumes are presented in Supplementary Fig. 5a). **b** B16.F10 tumor-bearing mice were treated as described in (**a**), with additional treatment group receiving three i.t. doses of 50µg smSTING agonist. Long-term survival is shown (**$P = 0.006$, ***$P = [0.0004–0.0006]$, Mantel–Cox log-rank comparisons for the indicated groups, see Supplementary Fig. 5d, e for tumor growth data). **c** A20 tumor-bearing mice were treated as described in (**a**) with saline or varying quantities of SYNB1891. Tumor growth data and ratio of C.R. are shown (**$P = 0.0014$ (blue stars—indicated group vs 10$^8$ CFUs SYNB1891), ***$P = 0.0004$ (pink stars—indicated group vs $5 \times 10^8$ CFUs SYNB1891) ****$P < 0.0001$ (orange stars—indicated group vs 10$^9$ CFUs SYNB1891) one-way ANOVA with Tukey's multiple comparisons tests for Day 21, see Supplementary Fig. 6a, b for individual tumor volumes and long-term survival). **d** A20 tumor-bearing mice were treated as described in (**a**) with saline or SYNB1891-cmR (SYNB1891 with chloramphenicol resistance gene). T cells were depleted by administration of anti-CD4 or anti-CD8 antibodies, isotype antibody injected as control. Long-term survival is shown. (*$P = 0.041$, ****$P < 0.0001$, N.S. = not significant, Mantel–Cox log-rank comparisons). **e** Mice treated as described in (**c**) and remained tumor free by Study day 50 were rechallenged by subcutaneous injection of A20 cells on the contralateral flank alongside naïve age-matched controls. Tumor growth data is shown (Individual tumor volumes are presented in Supplementary Fig. 6c). **a–e** $n = 10$ mice per group. **a, c, e** Mean and s.e.m. are shown. Data are representative of two (**b, d, e**) and three (**a, c**) independent experiments.

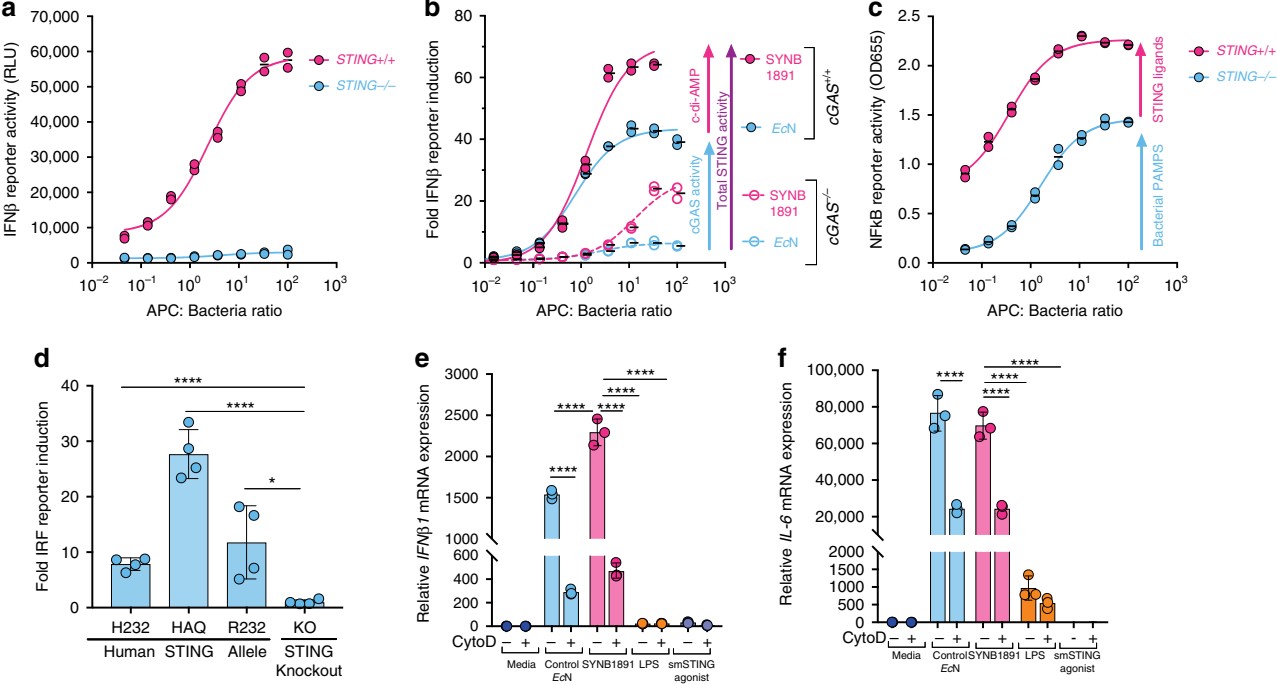

**Fig. 6 SYNB1891 activity in human antigen-presenting cells. a–d** THP-1 immortalized human monocyte cells containing an interferon-regulatory factor (IRF)-luciferase reporter and an NF-κB responsive colorimetric reporter were used. **a** THP-1 cells containing the endogenous STING allele (STING$^{+/+}$) or lacking STING gene (STING$^{-/-}$) were treated with pre-induced SYNB1891 at different ratios or media alone overnight. Cells were analyzed for the activation of the IRF reporter via luminescence with relative luminescence units (RLU). **b** Wildtype THP-1 cells (cGAS$^{+/+}$) and those lacking the *CGAS* gene (cGAS$^{-/-}$) were treated with pre-induced SYNB1891 or Control *Ec*N at different ratios, or media alone overnight. Cells were analyzed for the activation of the IRF reporter and fold IRF induction in treated cells relative to untreated media alone control is shown. **c**, STING$^{+/+}$ and STING$^{-/-}$ THP-1 cells were treated as described in (**a**) with pre-induced SYNB1891. Cells were analyzed for NFκB activity via colorimetric assay at OD655. **a–c** n = 2 biological replicates per group per APC: Bacterial ratio. **d** THP-1 cells containing an IRF-luciferase reporter and either the endogenous HAQ *TMEM173* (STING) allele, knock-ins of the H232 or R232 alleles or knockout of the *TMEM173* gene were treated with pre-induced SYNB1891 (MOI: 100) or media alone overnight. Cells were analyzed for the activation of the IRF reporter as described in (**b**) (*P = 0.0118, ****P < 0.0001, one-way ANOVA with Tukey's multiple comparisons tests, n = 4 biological replicates per group). **e, f** Monocyte-derived primary human DCs were treated with Control *Ec*N (MOI: 25), pre-induced SYN1891 (MOI: 25), LPS (100 ng/mL) or smSTING agonist (5 µg/mL) for 4 h. In indicated groups cells were pre-treated for 1 h with Cytochalasin D (10 µM). Human DCs incubated in media alone served as negative control. Cells were analyzed for the upregulation of *IFNB1* (**e**) and *IL6* (**f**) mRNA (****P < 0.0001, two-way ANOVA with Tukey's multiple comparisons tests, n = 3 biological replicates per group). **a–f** Data are representative of two independent experiments. **d–f** Data with mean and s.d. shown. Each circle represents an independent experimental replicate.

as T cells where STING activation could be detrimental to the formation of antitumor immunity[8,9]. In murine tumor models, SYNB1891 treatment resulted in efficacious anti-tumor immunity, long-term efficacy, and the establishment of immunological memory. Accordingly, while a recent study demonstrated that high doses of smSTING agonist can have inhibitory effects on the development of immunological memory[9], all A20-tumor bearing mice treated and cured by SYNB1891 were resistant to secondary tumor challenge, regardless of dose level. Beyond STING activation, SYNB1891 was found to trigger expression of additional proinflammatory cytokines which may result in superior long-term efficacy. Due to *Ec*N's intrinsic tumor tropism and serum sensitivity, intratumoral injection of this immunostimulatory vector results in a cytokine response concentrated to the TME. This is advantageous from a safety perspective to minimize systemic exposure, but also from a biological perspective to generate defined chemokine gradients to steer immune cell trafficking into the tumor. Finally, translation of results from mouse models to humans is often a great challenge. Immune reactivity of mouse versus human cells to the synthetic smSTING agonist DMXAA is one relevant example, with promising preclinical results in mouse models but a lack of activity as a human STING agonist[43]. Moreover, the *TMEM173* gene (encoding hSTING) presents in the human population in several common alleles with varying

binding properties to agonistic molecules[42]. SYNB1891 treatment induced type I IFN production in human dendritic cells and in human monocytes containing a variety of the most common human STING (*TMEM173*) gene variants. Important differences between murine and human systems were uncovered, including the contribution of cGAS towards STING activation following SYNB1891 treatment of human APCs.

Beyond the specific description of SYNB1891 and its mechanism of action in the context of immuno-oncology, our study demonstrates the value of synthetic biology to rationally design microorganisms to fight human disease. A variety of bacterial based therapies have previously been investigated for the treatment of cancer, with the majority using attenuated pathogens such as *Salmonella typhimurium*, *Clostridium novyi*, and *Listeria monocytogenes*[11]. *Listeria*-based approaches typically use attenuated strains as a vaccine vector, taking advantage of the intrinsic capacity of the bacterium to forcefully invade APCs and engineering expression of shared tumor-associated antigens or unique neoepitopes personalized to each patient[44–47]. Similar to traditional vaccine approaches, this requires the a priori identification of efficacious target antigens. While *Salmonella*- and *Clostridium*-based therapeutics also require the use of highly attenuated strains, such approaches primarily take advantage of the oncolytic potential of these pathogenic bacteria[48]. Clinically

these oncolytic strains have either failed to exhibit robust efficacy or have been hindered by dose-limiting toxicities[11]. In contrast, SYNB1891 uses the non-pathogenic strain *Ec*N, which has a potentially advantageous safety profile and significantly broader set of tools for genetic manipulation and engineering. Using synthetic biology techniques, we rationally designed an *Ec*N strain that expresses the STING-agonist CDA and, following intratumoral injection, delivers targeted activation to APCs in the TME in order to generate and support antitumor immune responses.

Since previous bacterial based therapies with compelling pre-clinical evidence have failed to demonstrate efficacy in patients, clinical testing of SYNB1891 is critical to assess the translatability of our findings to human disease. A Phase I clinical trial of SYNB1891 intratumoral injection in patients with percutaneous accessible advanced or metastatic malignancies has already been initiated (ClinicalTrials.gov Identifier: NCT04167137). While safety and tolerability will first be assessed for the percutaneous route of administration, future investigations with radiologically guided intratumoral injection would enable treatment of visceral lesions. Positive indications for safety and efficacy would provide support for the development of additional engineered strains with rationally designed functionalities tailored to the needs of specific cancer and patient subtypes. Although the application of synthetic biology to human therapeutics is still in its infancy, our work highlights its potential and provides guidance for the development of future approaches intended for clinical evaluation.

## Methods

### Data reporting
No statistical methods were used to predetermine sample size. Mice utilized for in vivo experiments were randomized based on tumor volume and investigators were not blinded to allocation during experiments or outcome assessments.

### Mice
SPF C57BL/6, BALB/c, STING$^{−/−}$ (C57BL/6J-*Tmem173*$^{gt}$/J) and TLR4$^{−/−}$ (B6.B10ScN-*Tlr4*$^{lps-del}$/JthJ) female mice were purchased from Jackson Laboratory. Animals were maintained under specific pathogen-free conditions in a climate controlled holding room at an ambient temperature of 21 ± 2 °C with relative humidity ranges between 30% and 70% and automated controlled 12 h dark/light cycle. Mice were used for the experiments at 7 to 9 weeks of age. Animal housing and all procedures related to in vivo experiments were reviewed and approved by Mispro's Institutional Animal Care and Use Committee (Mispro Biotech Services, 400 Technology Square, Cambridge, MA, 02139) in accordance with the Animal Welfare Act.

### Growth and pre-induction of strains in shake flasks
For preparation of cultures at small scale using flasks, cells were inoculated in 4 ml of 2-YT media containing appropriate antibiotics and/or supplements in a 14-ml culture tube at 37 °C with shaking (250 rpm) overnight. The next day, cell cultures were diluted 1:100 in 40 ml of fresh 2-YT media in 125-mL baffled flasks at 37 °C with shaking (250 rpm) for 2–3 h, with appropriate antibiotics and/or supplements. For gene circuit induction, depending on the promoter system, cells were induced for 4–5 h with the addition of anhydrotetracycline (Sigma; aTc; 100 ng/ml final concentration) or by placing flasks in an anaerobic chamber (Coy) supplying an atmosphere of 85% nitrogen, 10% carbon dioxide and 5% hydrogen. Following induction, cells were washed and spun down by centrifugation at 4000*g* for 10 min at 4 °C. Cell pellets were utilized to analyze for cyclic-di-AMP by LC-MS/MS (see, *Cyclic-di-AMP (CDA) quantification by LC-MS/MS*). If cell stocks needed to be stored for future usage, they were harvested by centrifugation, washed with cold PBS buffer, resuspended in formulation buffer containing 15% glycerol and PBS, and stored at −80 °C. All bacterial stocks utilized for in vitro or in vivo assays were first validated to contain >90% viability, purity as determined by growth in the presence/absence of antibiotics or metabolites associated with strain auxotrophy and for functionality looking for expression of payload following induction.

### Growth and induction of strains in bioreactors
Cells were inoculated in 50 mL of FM3 medium supplemented with diaminopimelic acid and thymidine in a 500-mL Ultra-Yield flask (Thomson). Cells were grown at 37 °C with shaking at 350 rpm overnight. Next day, 110 mL of the overnight culture was used to inoculate 4 L of FM3 in an Eppendorf BioFlow 115 bioreactor (starting OD600 of ~0.5). The fermenter was controlled at 60% dissolved oxygen with agitation, air, and oxygen supplementation, and controlled to pH 7 using ammonium hydroxide. Cells were harvested after 5 h and at OD600 ~30. Cells were harvested by centrifugation at

4500*g* for 30 min at 4 °C, resuspended in formulation buffer, and stored at −80 °C. OD600 was measured on a BioPhotometer Plus spectrophotometer (Eppendorf). Before use, frozen stocks were evaluated for viability (>90%), purity and functionality as described above.

### Bacterial strain construction
List of strains used in this publication provided in Supplementary Table 1. *Escherichia coli* Nissle 1917 (*Ec*N), designated as SYN001 here, was purchased from the German Collection of Microorganisms and Cell Cultures (DSMZ Braunschweig, *E. coli* DSM 6601). SYN001 with strep resistance was obtained by streaking ~ 10$^{11}$ cells of SYN001 on LB plate containing 300 ng/mL streptomycin and taking the single colony formed to have mutated to be strep resistant and designated as SYN094. Plasmid harboring the *luxABCDE* operon under the control of constitutive promoter was synthesized and subcloned into pST vector by Genewiz. Three different cyclic dinucleotide synthase genes *dncV*, *dacA*, and *cdaS* were codon optimized, synthesized by IDT and subcloned into a Synlogic vector containing kanamycin resistant gene, a medium copy number origin of replication p15A and under the regulatory control of the P$_{tet}$ promoter. Similarly, gene encoding GFP protein under the control of P$_{tet}$, P$_{sal}$ or P$_{cmt}$ promoters were synthesized by IDT or Genewiz and subcloning into the same Synlogic vector with p15A origin and kanamycin resistance. The gene encoding GFP protein under the control of P$_{fnrs}$[37,38] promoter was also subcloned into a Synlogic vector containing ampicillin resistant gene, a low copy number origin of replication pSC101 and constitutive promoter driven gene encoding mCherry fluorescent protein. These plasmids were used to electroporate SYN001 or SYN094 with integrated cassette into *malEK* locus to constitutively express mCherry protein. After the electroporation (Eporator, Eppendorf, 1.8-kV pulse, 1-mm gap length electro-cuvettes) the transformed cells were selected as colonies on LB agar (Sigma, L2897) containing proper antibiotics. For deletion of the *dapA* gene, *thyA* gene and phage fragment (Φ), a single round of PCR was performed using pKD3 or pKD4 as the template DNA[37,38]. The primers were designed to generate a dsDNA fragment that contained homology adjacent to the targeted gene locus or fragment in the *Ec*N chromosome and a chloramphenicol or kanamycin resistance gene flanked by frt sites. The resulting knockout fragment typically included ~40–60 base pairs of *Ec*N homology on its 5′ and 3′ end (DNA Sequences for Genomic Deletions). Parent strain, which contains the plasmid pKD46, was transformed with the knockout fragment by electroporation. Colonies were selected on LB agar containing proper antibiotics (30 ng/mL chloramphenicol or 100 ng/mL kanamycin) and proper auxotrophy supplements (diaminopimelate (100 μg/mL) and/or thymidine (3 mM) and correct recombination events were verified by PCR. A verified colony was saved for future operations. A plasmid containing constitutively expressed GFP protein was constructed similarly as described above into Synlogic vector with pSC101 origin and ampicillin resistance gene and transformed into the control Nissle strain with *dapA*, *thyA* and phage knockouts. Cyclic-di-AMP producing strain was created by insertion of genes into the *Ec*N chromosome. The intergenic locus exo/cea was identified as suitable integration site. Chromosomal insertions into the *Ec*N genome were carried out using the well-characterized lambda Red recombineering approach. For this insertion, (1) a pKD4-based plasmid containing ~1000 bp of 5′ and 3′ *Ec*N genome homology for recombination was built, followed by (2) insertion of the P$_{fnrs}$-*dacA* fragment into the plasmid by isothermal assembly (HiFi DNA Assembly Master Mix, NEB), (3) amplification of the insertion fragment from the plasmid by PCR (including *Ec*N homology regions and kanamycin resistance cassette, Q5 High-Fidelity Master Mix, NEB), (4) recombineering of the insertion fragment by electroporation via pKD46 and subsequent pKD46 removal, and (5) the removal of antibiotic resistance cassettes via pCP20 and subsequent pCP20 removal. All DNA sequences and detailed scheme for genomic insertion and deletions used in the construction of clinical candidate strain SYNB1891 are available upon request. Complete genome sequence of E.coli Nissle 1917 is listed in Genbank/EMBL/DDBJ accession number: CP007799. Plasmid maps and DNA sequences for CDA-producing enzymes are listed in the Supplementary Table 2 and Supplementary Fig. 8. The genome map, manipulated genome sequence of SYNB1891 and corresponding primers are available in Supplementary Fig. 9 and Supplementary Tables 3–7. Engineered strains described in this manuscript can be made available subject to a Material Transfer Agreement (MTA), which can be requested by contacting the corresponding authors. All requests will be reviewed by Synlogic to verify whether the request is subject to any intellectual property or confidentiality obligations.

### Antibiotic sensitivity test
Sensitivity of SYNB1891 to a panel of commonly utilized antibiotics was evaluated by Boston Analytical laboratory (Salem, NH) in accordance with Clinical and Laboratory Standard Institute (CLSI) Chapter M100 "Performance Standards for Antimicrobial Susceptibility Testing." Three antibiotic resistance trials were performed using an independent organism preparation and assay set up. SYNB1891 was grown on LB with 100 μg/mL DAP and 10 mM Thymidine agar and a suspension was prepared then adjusted to a known concentration using the spectrophotometer. Lawn plates were prepared using the adjusted concentration. A total of 16 Antibiotic disks were assessed. For each disk tested, a lawn plate was prepared by inoculating the surface of an LB/DAP/Thymidine agar plate with 200 μL of the adjusted suspension. Each disk was aseptically placed in the center of the plate. Sample plates were incubated at 30–35 °C for 18–24 h. A negative control was prepared by placing a blank disk on an inoculated

plate. Following incubation, the zones of inhibition were measured using calibrated calipers. Each unrounded result was documented in millimeters.

**Anerobic in vitro circuit induction**. The cryovials were thawed on ice and mixed well before measuring OD density. $10^{10}$ cells were taken into 1 mL PBS in a deep well plate, mixed by pipetting (duplicate each sample). Plate was spun down at 2750$g$, 4 °C for 10 min. Media was discarded and the plate with pellets were stored at −80 °C as time zero control samples. For induction of the circuit to produce cyclic-di-AMP, 500 μL culture from the thawed cryovial was added into 3.5 mL minimum media ($1 \times$ M9 + 50 mM MOPS + 2% glucose + 100 ng/mL diamino-pimelic acid + 3 mM thymidine) and recovered by shaking at 37 °C, 250 rpm for 30 min. The culture was then moved into anaerobic chamber to incubate statically for 2 h, at 37 °C. When finishing incubation, the tubes were brought out, mixed by vertexing and OD density was measured. Similar to time zero control samples, $10^{10}$ cells were taking into 1 mL PBS in a deep well plate, mixed well by pipetting (duplicate each sample), pelleted by centrifugation and stored at −80 °C. Samples were then sent for LC-MS/MS analysis of cyclic-di-AMP (see, *Cyclic-di-AMP (CDA) quantification by LC-MS/MS*). For the further in vitro experiments, induced bacteria were washed twice in sterile PBS and bacterial numbers were counted by Cellometer X2 (Nexcelom Bioscience).

**Subcutaneous tumor models and in vivo studies**. B16.F10 melanoma (ATCC, CRL-6475), EL4 T-cell lymphoma (ATCC, TIB-39), A20 B-cell lymphoma (ATCC, TIB-208), 4T1 mammary carcinoma (ATCC, CRL-2539) and CT26 colon carci-noma (ATCC, CRL-2638) cells were cultured under standard conditions as spe-cified by ATCC (37 °C incubator, 5% CO$_2$, humidified) in their recommended media formulations. Adherent cell lines were harvested following trypsinization. All cell lines were rinsed in Dulbecco's phosphate-buffered (DPBS) saline twice to remove excess FBS and resuspended in DPBS for implantation. The following number of cells were implanted for the indicated cell lines: B16.F10 ($2 \times 10^5$), EL4 ($5 \times 10^5$), A20 ($1 \times 10^6$), 4T1 ($1 \times 10^5$) and CT26 ($1 \times 10^6$). Mice were injected subcutaneously using a 26-gauge needle into the shaved, left or right flank and tumors were allowed to establish until they reached between 40 and 300 mm³, as indicated for each experiment. Mice were then randomized into treatment groups based on tumor volume in order to create groups with approximately equal average tumor volumes. Intratumoral injections involved direct insertion of the needle into the largest section of the tumor and the slow, consistent release of contents. Intravenous injections were performed via the tail vein. For the efficacy studies tumor-bearing mice were treated on Study days 1, 4, and 7. As indicated, 50 μL of either bacteria, smSTING agonist or saline (as injection control) were injected into tumors. Tumor volume ($V$) was measured via electronic caliper where $V =$ length $\times$ width² $\times$ 0.5. Mice having tumors exceeding 2000 mm³ were euthanized per IACUC protocol. Unless otherwise stated, once 50% or more of mice in an experimental group were removed from study no further tumor measurements were taken and all remaining mice in said group were euthanized, with the exception of any mice having no palpable tumor mass. All complete responder (C.R.) mice, or those having no palpable tumor by the indicated timepoint, were monitored until at least study day 60 and at least 50 days post the final treatment dose to ensure no tumor relapse was observed. In some studies T cells were depleted by intraperitoneal injection of 200 μg of antibody resuspended in DPBS (final volume 100 μl) 1 day before bacterial treatment initiation and twice weekly throughout the study with an isotype matched antibody injected as control using the following antibodies: rat IgG2b isotype control (clone: LTF-2, BioXcell, BP0090), anti CD4 (clone: GK1.5, BioXcell, BP0003-1), or anti CD8α (clone: 2.43, BioXcell, BP0061).

**Tumor homogenates**. At the indicated time points mice were euthanized and tumors were removed via dissection. Harvested tumors were placed in sterile beadbug homogenizer tubes prefilled with 2.8 mm stainless steel beads (Sigma, Z763829) containing 500 μl sterile DPBS, weighed and homogenized for 5 min utilizing a Fast Prep-24 homogenizer (MP Biomedicals). Whole tumor homo-genates were utilized for the quantification of cyclic-di-AMP and bacterial quan-tification (see below). Tumor supernatants were utilized for the analysis of intratumoral cytokines and were prepared as follows. Triton-X100 (Thermo Sci-entific, BP151-100) was diluted in DPBS to generate a 10% Triton-X100 stock solution and added to tumor homogenates to make a final concentration of 1% Triton-X100. Treated homogenates were pipetted vigorously and vortexed to mix and centrifuged at 10,000$g$ for 10 min. Tumor supernatants were then collected, transferred to a 96-well plate, treated with HALT protease inhibitor (Thermo Scientific, 1861278), frozen and stored at −80 °C to allow for quantification of all samples and time points in the same run.

**Bacterial quantification**. To perform colony unit forming (CFUs) assays, whole tumor homogenates or blood collected by cardiac puncture and treated with EDTA were transferred to 96-well plates, 10x serial diluted into DPBS and 10 μl of diluted samples were streaked across LB-agarose plates. Depending on the antibiotic resistances or auxotrophies of the bacterial strain being quantified, LB-agarose plates were supplemented with 3 mM thymidine (Sigma, T1895), 100 μg/mL dia-minopimelic acid (Sigma, D1377), 30 μg/mL streptomycin (Sigma, S9137-100G) or

300 μg/mL chloramphenicol (Sigma, C0378-100G). Plates were incubated at 37 °C overnight and the number of colonies formed was quantified from the lowest dilution where 30 colonies or greater were present. CFU bacterial quantification exhibited a limit of detection of ~100 CFUs per milliliter of sample with a limit of quantification of ~3000 CFUs per milliliter. Relative bioluminescence from 100 μL tumor homogenate in a 96-well opaque plate, compared to background from saline treated control, was measured using the Synergy Neo Microplate Reader (BioTek Instruments). To detect bacterial luminescence for excised tumor and/or sur-rounding tissue, samples were placed in a G:Box (Syngene) gel doc system. Images under white light were captured to delineate the border between the B16F10 tumors and the surround healthy subcutaneous skin tissue. Bacterial luminescence was capture in the absence of light with full aperture and a 2 min exposure time. For the detection of SYNB1891 bacterial DNA in blood and tumor a SYNB1891 probe specific qPCR assay was developed and validated using good laboratory practices standards at QPS, LLC, Newark DE, USA. Tumors and EDTA treated blood were snap frozen and shipped to QPS for analysis. The qPCR analysis was performed using customized SYNB1891 forward, reverse primers and probe (Life Technologies, ThermoFisher Scientific) according to analytical procedure (method)[7] number 1813-1804. SYNB1891 primer/probe specific sequences were as follows: Forward primer "5′- AGGATACTCATGTTGGAAAAGTCCAT-3′"; Reverse primer "5′- CCCGCATCCCACGCTAAC-3′"; Probe "5′-6FAM CTTGCCCCCCTTATATMGBNFQ-3′". The qPCR detection limit was 10 copies of SYNB1891 DNA per 1 μg of total DNA with a quantification limit of 25 copies per sample.

**Cytokine quantification**. Tumors were harvested and tumor supernatants were isolated as described above. Blood was collected by cardiac puncture and was transferred to a serum separator tube containing EDTA (BD, 365967). Serum was isolated by allowing the blood to clot for 30 min at room temperature followed by centrifugation for 90 s at 15,000$g$. Serum layer was transferred into a 96-well plate, treated with HALT protease inhibitor (Thermo Scientific, 1861278) and stored at −80 °C until analysis. IL-6, TNFα, IL-1β, MCP-1, IL-2, Granzyme B and IFN-γ from tumor supernatants and serum level of IL-6 and TNFα were quantified using a custom MILLIPLEX mouse cytokine magnetic bead panel kit (MILLIPLEX Multiplex Assay, Millipore Sigma) following manufacturer's instructions and analyzed on a MAGPIX analyzer (Luminex). IFNβ was quantified from culture media and tumors using the VeriKine mouse interferon beta ELISA kit (PBL) following manufacturer's instructions. IL-6 and TNFα were quantified from culture media using mouse Quantikine ELISA kits (R&D systems) following manu-facturer's instructions. All ELISAs were run on a 96-well plate and analyzed via colorimetric readout using the Synergy Neo Microplate Reader (BioTek Instru-ments). Intratumoral levels of IFNα, IL-6, GM-CSF and IL-15 were quantified using Biolegend's Mouse Inflammation LEGENDplex Custom flow analyte kit (Biolegend) following manufacturer's instructions and were analyzed on a Cytoflex LX flow cytometer (Bechman Coutler).

**Evaluation of inducible promoters in vivo**. B16.F10 tumors were implanted and allowed to reach a volume of 100–200 mm³. Mice were randomized based on tumor volume into various experimental groups and intratumorally injected with either saline or $5 \times 10^5$ CFUs of bacteria constitutively expressing mCherry (RFP) and containing a green fluorescence protein (GFP) cassette under the control of an inducible promoter. 48 h post-bacterial injection mice were administered with either PBS (as control), 10 μg anhydrous tetracycline (aTc), 6 mg $p$-iso-propylbenzoate (Sigma; cumate, cmt) or 100 mM sodium salicylate (Sigma; aspirin, sal) via intraperitoneal injection. Mice were euthanized 1 or 16 h post-induction, tumors were harvested by dissection and placed into single cell suspension by mechanical disruption though a 40 μM cell strainer. Samples were placed in a 96-well round bottom plate and analyzed on a Macsquant VYB flow cytometer (Miltenyi Biotec). Bacteria were differentiated from mouse cells via size and expression of RFP, and the percentage of induced cells (GFP$^+$) was quantified. Flow cytometer settings and gating strategy are shown in Supplementary Fig. 10.

**Cyclic-di-AMP (CDA) quantification by LC-MS/MS**. To quantify CDA pro-duction in vitro, bacteria were grown and induced in an anaerobic chamber as described above (See, *Growth and pre-induction of strains in shake flasks*). Bacterial cells were harvested by centrifugation, rinsed twice with DPBS, counted using a Cellometer X2 (Nexcelom), $10^{10}$ cells transferred to a 96-well plate and frozen at −80 °C. Tumors were harvested and homogenized as described above. Tumor homogenates were transferred to a 96-well plate, frozen and stored at −80 °C. CDA was then quantified from bacterial pellets or tumor homogenates by LC-MS/MS. In brief, tumor homogenate was extracted with 9 parts methanol, vortexed, and centrifuged at 2300$g$ for 5 min at 4 °C. Supernatants were diluted 10-fold with 0.1% formic acid prior to analysis. CDA was separated and detected on a Vanquish UHPLC/TSQ Altis LC-MS/MS system (Thermo Fisher Scientific). Samples were injected (1 μL) and separated on an Ace Excel 2 μm 100 Å C18-AR 100 × 2.1 mm column (Advanced Chromatography Technologies Ltd.) and analyte eluted using a linear gradient of 0 to 98% B mixed with A from 1 to 3 min (A: 0.1% formic acid; B: acetonitrile, 0.1% FA) at 0.6 mL/min at 50 °C. CDA was detected using selected reaction monitoring (SRM) of a compound specific mass transition 659.1 > 136 m/z

in positive electrospray ionization mode. Peaks were integrated and peak areas used to calculate concentrations of the unknowns using a 1/X weighted linear 7-point standard curve from 0.032 to 250 ng/mL using Xcalibur Quan Browser (Thermo Fisher Scientific).

**Human serum (complement) sensitivity.** Fresh healthy human serum samples from three donors were obtained from Research Blood Components LLC, Watertown, MA. Aliquots of serum from each donor were heat inactivated at 56 °C for 30 min (after aliquots reached the temperature). Induced SYNB1891-cmR bacteria (see *Anerobic in vitro circuit induction*) were resuspended at a concentration $1 \times 10^7$ cells/ml and 50 µl of bacteria were added to 450 µl of human serum (untreated or heat inactivated). Next, bacteria were incubated at 37 °C with shaking for $T = 0$, 45, 90, and 180 min and then plated in serial dilutions for CFU counting on LB-agarose plates supplemented with thymidine, diaminopimelic acid and chloramphenicol (see *Bacterial quantification from* in vivo *samples*) to determine susceptibility to serum lysis.

**Macrophage and dendritic cell cultures.** Murine RAW264.7 macrophage cell line (ATCC) was maintained according to ATCC's recommendations in DMEM-GlutaMAX media containing 10% FBS and penicillin-streptomycin (100 IU/ml and 100 mg/ml). For generation of murine bone marrow-derived dendritic cells, bone marrow cells were sterilely harvested from the tibia and femur, red blood cell lysed using sterile Red Blood Cell Lysing Buffer Hybri-Max (Sigma) and frozen in FBS containing 10% DMSO (Sigma) at −80 °C. Bone marrow cells were thawed, rinsed to remove DMSO and resuspended at a concentration of $5 \times 10^5$ cells/mL in culture media RPMI 1640-GlutaMax containing 5% FBS, 50uM 2-mercaptoethanol, penicillin-streptomycin (100 IU/ml and 100 mg/ml) and 20 ng/ml GM-CSF (BioLegend). After day 3, all non-adherent cells were removed, and adherent dendritic cell clusters were cultured in the fresh medium for another 6 days before use. Every other day half of the medium was replaced with fresh culture medium.

**Microscopy.** For microscopic imaging of BMDCs, cells were plated on poly-D-L-lysine covered glass coverslips in 24 well plate at concentration of $5 \times 10^5$ cells/well in RPMI 1640-GlutaMax media containing 5% FBS without antibiotics. After cells adhered, in some wells cells were pre-treated with Cytochalasin D (10 µM) for 1 h. Then live pre-induced SYNB1891-*gfp* bacteria at MOI of 25 were added to BMDCs. Plates were spun at 350g for 2 min to allow bacteria to come in contact with DCs and incubated at 37 °C for 1 h to allow phagocytosis. After 1 h non-internalized bacteria were washed out by rinsing three times with ice-cold DPBS and cells were fixed for 20 min at RT in 4% PFA in PBS. Cells were rinsed 2-3 times with 1 mL PBS (1 min each) and permeabilized for 10 min with permeabilization buffer (0.2% saponin in PBS, 0.03 M sucrose, 1% BSA). Then cells were treated for 60 min with blocking buffer (2% goat serum, 1% BSA, 0.1% cold fish skin gelatin, 0.1% saponin and 0.05% Tween-20 in PBS, pH7.2) and stained with primary antibody against GFP (rabbit anti-GFP, Abcam, Cat. # ab6556) and LAMP-1 (rat anti-mouse-LAMP-1 (1D4B), eBioscience, Cat. # 14-1071-82) at 1:250 dilution in dilution buffer (PBS, 0.05% Tween-20, 1%BSA, 0.1% saponin) overnight at 4 °C. Next day cells were washed 3 times, 5 min each, in 1 ml of dilution buffer and stained with secondary antibody donkey anti-rabbit IgG (Alexa488, ThermoFisher Scientific,. # A-21206) and goat anti-rat IgG (Alexa 647, ThermoFisher Scientific, # A-21247) in dilution buffer for 60 min at RT at dilution 1:750. Next, BMDCs were washed 3 times, 5 min each, in 1 mL of dilution buffer, cell nuclei and F-actin were stained with NucBlue and ActinRed 555 Probes (ThermoFisher Scientific) according to the manufacturer's protocol. After staining, BMDCs were rinsed two times with DPBS, coverslips were mount with ProLong® Gold Antifade Mountant (ThermoFisher Scientific) and dried at RT overnight. Cells were imaged using 40 × (aperture 0.75) and 60 × (aperture 1.42) objectives with EVOS™ FL Auto 2 Imaging System (Invitrogen). Post-acquisition contrast adjustment was performed using the ImageJ software. A representative example for the quantification of bacteria co-localized within dendritic cells is shown in Supplementary Fig. 11. The average number of bacteria per dendritic cell in a field of view (FOV) was quantified and shown as the ratio of bacteria internalized by dendritic cells to the total number of dendritic cells per FOV.

**Cytokine responses in Macrophages and Dendritic Cells.** RAW264.7 macrophages or BMDCs were seeded in 24 well plate at concentration of $5 \times 10^5$ cells/well in 0.5 mL of corresponding media (see *Macrophage and dendritic cell cultures*) without growth cytokines and antibiotics. Human monocyte-derived primary DCs (StemCell Technologies, 70041) were defrosted and prepared for downstream application according to the manufacturer's protocol. Human DCs were seeded in 96-well plate at concentration of $1 \times 10^5$ cells/well in 0.2 mL of RPMI 1640-GlutaMax media containing 10% FBS. Two hours later after cells adhered, cells were stimulated with the following treatments in triplicate: live control bacteria *Ec*N or live induced SYNB1891 bacteria at MOI of 25; ultrapure LPS (Invivogen) at concentration of 100 ng/mL; stable smSTING agonist (2'3'-c-di-AM(PS)$_2$ (Rp,Rp), Invivogen) at concentration of 5 or 25 µg/mL. Plates were spun at 350g for 2 min to allow bacteria to come in contact with DCs or macrophages and incubated at 37 ºC for 1 h to allow internalization of treatments. After 1 h non-internalized bacteria, LPS or smSTING agonists were washed out two times with warm DPBS, fresh

media containing antibiotics were added and cells were incubated at 37 ºC. Depending on the readout, cells or culture media were harvested for analysis at 2, 4, 6, or 18 h post-treatment. Macrophages and DCs incubated in media only were utilized as a negative control. Cell supernatants and cells were collected and analyzed for the presence of cytokine proteins and gene expression. Quantikine ELISA kits (R&D Systems) were used to detect mouse cytokines IL-6 and TNF-α via standard colorimetric readout or custom MILLIPLEX mouse cytokine magnetic bead panel kits (MILLIPLEX Multiplex Assay, Millipore Sigma) were used to detect IL-6, IL-1β, and IL-12p70 and analyzed on a MAGPIX analyzer (Luminex). PBL AssayScience ELISA was used to detect mouse IFN-β1 via standard colorimetric readout. In some experiments cells were pre-treated for 1 h with Cytochalasin D (10 µM in corresponding media) before stimulation to stop phagocytosis. Cytochalasin D was present in the media during first hour of stimulation until wash.

**RNA isolation, cDNA preparation, and qPCR analysis.** Cellular RNA was isolated with TurboCapture 96 mRNA Kit (Qiagen) according to the manufacturer's protocol. For cDNA preparation, cDNA was synthesized from isolated RNA using High-Capacity cDNA Reverse Transcription Kit with RNase Inhibitor (Thermo Fisher Scientific) according to manufacturer's protocol. For 20 µl reaction, 2X RT master mix was prepared as follows: 10X RT Buffer 2 µL, 25X dNTP mix 0.8 µL,10X Random primers 2 µL, MultiScribe RT (50 U/µl) 1 µL, RNAase inhibitor 1 µL, Nuclease-free H2O 3.2 µL, and mRNA 10 µL. PCR plate was run 10 min at 25 °C and 120 min at 37 °C in T100 Thermal Cycler (Bio-Rad). For the quantification of mouse *Ifnb1* (Mm00439552_s1 Taqman probe), mouse *Il6* (Mm00446190_m1), mouse *Il12a* (Mm00434169_m1), mouse *cd86* (Mm00444540_m1), human *IFNB1* (Hs01077958_s1), human *IL6* (Hs00174131_m1), human *TNF* (Hs00174128_m1), human *CD86* (Hs01567026_m1), human *CD40* (Hs01002915_g1), human *IL-12A* (Hs01073447_m1), and human *IL-1B* (Hs01555410_m1) mRNA expression, a 10 µL qPCR reaction was performed using Taqman assay probes and TaqMan Fast Advanced Master Mix, containing the following components: 2X Master Mix 5 µL, Taqman gene probe (20×) 0.5 µL (FAM-labeled), Taqman mouse or human β-actin (endogenous control)* probe (20×) 0.5 µL (VIC-labeled), cDNA 4 µL. Reaction was run in MicroAmp™ Optical 384-Well Reaction Plate 20 s at 95 °C and 40 cycles (1 s—95 °C, 20 s—60 °C) in QuantStudio 6 Real Time PCR System (Applied Biosystems). Gene expression was analyzed by using comparative $C_T$ Method ($\Delta\Delta C_T$ Method). The Prism7.0 c software (GraphPad Software, San Diego, CA) was used for all statistical analysis.

**Human monocyte THP-1 reporter cell line assay.** THP1-Dual cells (thpd-nfis) containing an IRF-luciferase reporter and NF-κB-alkaline phosphatase (SEAP) reporter, THP1-Dual KO-cGAS (thpd-kocgas) cells with the stable knockout of the cGAS gene and THP1-Dual KO-STING (thpd-kostg) cells with the stable knockout of the STING gene were purchased from InVivoGen and cultured following the manufacturers protocols. THP-1 cells lines containing the following TMEM173 (STING) alleles were utilized: H232 (InVivoGen, thpd-h232), HAQ-THP1 endogenous allele (InVivoGen, thpd-nfis) and R232 (InVivoGen, thpd-r232). Cells were harvested, pelleted and resuspended in 50% fresh RPMI 1640-GlutaMax media containing 10% FBS (RPMI-10) and 50% supernatant from expansion cell culture, and $1 \times 10^5$ cells in 225 µL per well were transferred to a 96-well cell culture plate. Bacteria were pre-induced to express CDA (as described above), rinsed three times in cold DPBS and resuspended in RPMI-10. Bacteria (at the indicated ratios) or media alone were added to the THP-1 cells to reach a total volume of 250 µL. Cells were incubated overnight, pelleted and supernatant was transferred to a 96-well white opaque plate for luciferase or 96-well transparent plate for phasphotase activity detection. Quanti-Luc (InVivoGen, rep-qlc1) or Quanti-Blue (InVivoGen, rep-qbs1) solutions were reconstituted and utilized following manufacturers protocol to assay luciferase or phosphatase activity of THP-1 culture media. Luminescence or optical density signal was measured using the Synergy Neo Microplate Reader (BioTek Instruments). The Prism7.0 c software (GraphPad Software, San Diego, CA) was used for all statistical analysis.

**Data analysis.** The Prism7.0 c software (GraphPad Software, San Diego, CA) was used for all statistical analysis. The details of the statistical tests carried out are indicated in the respective figure legends. Values were compared using either a Student's *t* test to compare two groups or one-way ANOVA with Tukey's multiple comparisons posttests to compare the variance in the multiple group means with one independent factor (e.g. treatment group). We employed two-way ANOVA with Tukey's multiple comparisons posttests when we had the effect of two factors (e.g. treatment and time (or genotype)) on a dependent variable. In in vivo experimental cases where some mice were removed from study for a treatment group upon reaching a study endpoint, we performed comparisons between the groups using one-way ANOVA with Tukey's multiple comparisons posttests at the indicated timepoint. For Kaplan–Meier survival experiments, we performed a log-rank (Mantel-Cox) test.

**Reporting summary.** Further information on research design is available in the Nature Research Reporting Summary linked to this article.

## Data availability

Complete genome sequence of E.coli Nissle 1917 is listed in Genbank/EMBL/DDBJ accession number: CP007799. Plasmid maps and DNA sequences for CDA-producing enzymes are listed in the Supplementary Table 2 and Supplementary Fig. 8. The genome map, manipulated genome sequence of SYNB1891 and corresponding primers are available in Supplementary Fig. 9 and Supplementary Tables 3–7. Engineered strains described in this manuscript can be made available subject to a Material Transfer Agreement (MTA), which can be requested by contacting the corresponding authors. All requests will be reviewed by Synlogic to verify whether the request is subject to any intellectual property or confidentiality obligations. Additional data underlying the figures and supplementary information are available from the corresponding authors on reasonable request.

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

## Acknowledgements

Drs. James J. Collins, Roger Geiger, Filip Janku, Timothy K. Lu, Neil Segal, Dmitriy Zamarin and Weiping Zou for discussions and comments on the manuscript. Megan McPhillips, Makswell Vittands, Nate Scaffidi and Alice Gao for technical assistance. Timothy Mayville, Scott Hamilton, Michael Jessel and Justin Wong for bacterial fermentation process development and material generation.

## Author contributions

J.M.L. conceived and supervised the project. D.S.L., A.S., and J.M.L. planned experiments, analyzed data and wrote the paper with assistance from K.A.W.; N.L., P.F.M., A.F., and J.G. designed and performed bacterial engineering and in vitro bacterial analyses. A.S., C.P., and C.G. designed and performed in vitro immunological assays and analyzed the data. S.K., R.C., and D.L. performed in vivo assays. K.A.W. supervised in vivo assays and analyzed data. M.J. performed mass spectrometry analyses. C.B., M.M., and A.A-F. devised fermentation conditions and provided essential materials.

## Competing interests

All authors are or were employees of Synlogic, Inc.
