## [Peer Review File · Nature Communications]

Reviewers' Comments:

Reviewer #1:

Remarks to the Author:

In their manuscript, Leventhal et al. describe a novel constructed strain of probiotic *E. coli* Nissle 1917 for tumor therapy. This is very important work since novel immunotherapy strategies are required to fight this devastating disease. The authors introduce into the bacteria the ability to synthesize dicyclicAMP a second messenger mainly used by Gram-positive bacteria which is known to activate in mammalian cells intracellular signaling via the STING pathway. In addition, they introduce two attenuations of the bacteria with the aim that this strain will be usable in clinical trials. By extensive in vitro using murine and human lines as well as by in vivo testing in mice they are able to show the safety and efficacy of the bacteria against several transplantable tumors. In addition, they provide evidence that the treatment of tumors in mice elicits and CD8 killer T cell response responsible for the tumor clearance in some of the animals.

This is a very intensive and interesting study and needs to be presented to the general readership of Nature communications. Especially interesting is the fact that the authors restrict their treatment to intratumoral application. The bacteria were constructed to address the antigen presenting cells that activate the anti-tumor T cell response. Hardly any bacteria are found outside the tumors. This implies that the activation of such T cells takes place in the tumor itself and not in the draining lymph node. This could be a very important finding and should be discussed in the manuscript. However, before accepting this manuscript for publication that a few points need to be addressed. It presently feels rather incomplete.

1. The title is very unfortunate and does not provide information on the target of their synthetic biology attempt. Please specify.
2. Line 50: TME as abbreviation should be introduced in the introduction when the term tumor microenvironment is used the first time.
3. Line 70: An extensive study on probiotic *E. coli* including Nissle as tumor therapy was published and is not quoted here (Kocijancic et al. Oncotarget 2016).
4. Line 78: Histology and immunohistology would be appropriate here to learn about the localization of the bacteria in the tumor.
5. Line 122: is 8 days correct?
6. Line 137: The nitrate reductase promoter was described by Leschner et al. (Nucleic Acid Res. 2012) as tumor specific. This should be quoted.
7. Line 155: the authors claim that their bacteria containing a double attenuation should be unable to proliferate in tumors. They have not formally proofed that. In our hands Δ asd still have a limited capacity to survive and proliferate in mice. A more cautious statement should be made here.
8. Line 164: The author claim to have used APC but really employed RAW264.7.
9. Lines 200 and following. Very important finding that phagocytosis is required for induction of IFN β by SYN1891.
10. Line 225: PCR is used to detected SYN1891 in blood. It would be interesting whether they are still viable and can be determined as CFU (see comment 7).
11. Line 230: Several animals cleared B16 already after a single application of SYN1891. This is a very important finding. Very few attempts using bacteria were able to clear the B16 tumor under these conditions. Why were these animals not used for the tumor re-challenge? This is very important for generalizing the results.
12. Line 272: The authors claim a durable anti-response in multiple murine tumor while they nly showed it for two.
13. M&M and discussion: A very extensive study on direct application of therapeutic bacteria into the tumor has been published by Kocijancic et al. (Oncotarget 2017). The authors should discuss this and refer to it.
14. Line 342: the authors refer to serum sensitivity of ECN. Actually ECN is very resistant to human serum. However, the attenuations especially the deletion of asd might alter this dramatically. Therefore resistance of SYN1891 a gainst serum of different donor should be tested.
15. Discussion: Throughout their work, the authors use intratumor injection to apply their therapeutic bacteria. This is very much in accordance to the application of Coley's toxin because he

usually treated skin tumors. The authors largely ignore the need to apply this therapy also to deeper tumors. This needs to be discussed.

16. Discussion: In their in vitro experiments, the authors show that phagocytosis is essential for SYN1891 to elicit positive immune reactions. However, thus far it is clear from many studies that bacteria in the tumor are never found intracellular. This was also shown for different probiotic *E. coli* (Kocijajic et al. Oncotarget 2016). How do the authors reconcile these data with their findings on the requirement of phagocytosis.

17. Figure 3: The micrograph in Fig. 3e are of extremely low quality. Better quality pictures should be provided.

18. Figure 4f-g: TNF α , IFN γ and IL-1 β should be measured and shown.

19. Figure 5: Correct overlaps in Fig. 5a, c and d.

20. Supplementary Fig. 1a: Nothing can be recognized here. Provide better pictures.

21. Supplementary Fig. 6b-d: Provide data on TNF α and IL-1 β .

Reviewer #2:

Remarks to the Author:

In this manuscript, Leventhal and colleagues describe the construction and testing of an engineered bacterial strain, SYN1891, which expresses cyclic-di-AMP (CDA) from an *E. coli* Nissle 1917 chassis. They demonstrate the progressive development of a CDA expression cassette along with various attenuations that make the strain compatible for future human delivery. The authors then test the final version of the strain, showing functionality as a STING agonist in in vitro cultures of mouse-derived cells, in mouse tumor models and in in vitro cultures of primary human DCs.

The concept of engineering bacteria as cancer therapeutics, including strains aiming to generate anti-tumor immunity, has been developed in several contexts over the past decades – including several clinical trials. Nevertheless, this work represents one of the more comprehensive examples of preclinical testing of synthetically engineered bacteria in the context of therapeutic delivery to date. The manuscript is broadly well written and well described (with a few exceptions mentioned below) and presents a strong case for further testing of CDA expressing *E. coli* to target tumors via the STING pathway. The work will likely be of interest in the context of immunotherapy and synthetic biology.

Although I am excited for the potential of this work, several aspects of the data analysis, presentation and interpretation need to be addressed before the work can be recommended for publication.

1 – Data analysis and experimental detail provided

Throughout the manuscript the level of detail provided regarding individual experiments and their data analysis is insufficient to properly judge the work. In particular:

- a. the actual number of replicates needs to be provided for each experiment/group
- b. where multiple groups are being compared there is currently no indication that the statistics provided take into account an appropriate multiple comparisons adjustment. The statistics must take this into account so if they currently do not the experiments should be re-analyzed, and the details of the tests used along with all comparisons made should be provided in each case.
- c. Levels of sensitivity should be indicated in relevant tests through the work. For example, the lack of detection of live bacteria or bacterial DNA in blood is only relevant in the context of the unstated sensitivity of the tests used.
- d. With particular reference to the in vivo mouse tumor model experiments, the experimental designs were hard to follow. Readers would benefit considerably from schematics being added to the relevant figures to clearly demonstrate the numbers and groups treated alongside each

experiment.

e. The imaging analysis in Supplementary Fig 1a needs more clarity on how it was analyzed and the implications of the outcomes.

2 – Overstatement of individual results.

a. The number of samples tested in many of the individual experiments/timepoints is relatively low. While the story as a whole is convincing based on its overlapping lines of evidence, in several cases the individual experiments undertaken have low n combined with high variation and thus the authors need to take care not to overstate individual results and their interpretations. Eg.

i) it is hard to conclude anything from the single image panels in Fig 3 d-f, particularly with no details regarding how a bacterium was determined as being “associated” with a cell or not (a simple contrast adjustment on the images suggests that some of the bacteria with no arrows could be associated with cells. Further, what do the arrows mean?). Either counts of multiple fields of view and quantitative information on the analysis criteria should be provided to support the authors claims that CytoD “significantly inhibited the number of bacterial cells observed” or the claims should be far more circumspect in the text.

ii) The data relating to late t-cell response in the tumor form Fig 2/Supp Fig 2, shows a couple of examples of high response in some EcN mice, even at day 2, which would seem to go against the authors’ hypothesis. The variability of this data should be reflected more clearly in the text.

b. in the absence of blinding of the treatment and control groups, the reliance on subjective measures for the total clearance of tumors (combined with no information provided in the methods other than “mice having no visible tumor on dissection”) makes these measures hard to interpret, particularly with low numbers of mice. Again, this should be reflected more clearly in the text.

3 – Lack of display of individual datapoints

With the number of samples used throughout the study, all datapoints should be displayed individually on graphs - in every case. This is currently not the case in several panels – (it appears that in fig 2b, for example, an n=2 dataset is being displayed as a mean +/- STD). Similarly, all tumor weight and volume graphs and some metabolite quantification graphs need to have individual datapoints displayed.

4 – Off target testing

Throughout the manuscript there are several references to off target effects of current STING agonists, with the suggestion that targeting through the use of EcN may overcome this issue. While the results of CytoD treatment to prevent phagocytosis by phagocytic cells are shown, the conclusions would be strengthened if the effect of EcN on off target cells was tested directly to substantiate these claims.

5 – Lack of sequence and resource sharing detail

While it is recognized that the authors have stated that “all DNA sequences and detailed schemes for genomic insertion and deletions used” are available upon request, there is no reason that these details should not be included directly within the supplementary materials at publication (or full genome sequences deposited in an open-source sequence repository), as is routine practice for synthetic biology publications.

Similarly, there is no mention of the availability and sharing of the strains, which should be made available for example by deposit with addgene.

These aspects are critical to allow other researchers in the field to reproduce the work.

6 – Discussion of the work in the broader context of therapeutic bacterial delivery.

While there is a lot of attention given to the advances this approach may give over alternative available STING agonists, the authors need to more extensively cite and discuss previous examples of anti-tumor immunity generation using engineered bacteria and how this study fits within that context. In particular, several studies have utilized attenuated *Listeria monocytogenes*, which also target phagocytic cells and naturally secrete CDA to activate STING pathways (indeed the *dacA* gene used in this study is from *Listeria monocytogenes*), leading to the generation of anti-tumor immunity (eg. Gunn et al, *J Immunol* 2001; Singh et al, *J Immunol* 2005; Shahabi et al, *Immunother*, 2008). These warrant further discussion.

Additionally, in light of the previous lack of strong efficacy for bacterial anticancer therapeutics in the clinic to date the authors should discuss potential shortfalls of the approach. As one example, the amount of bacteria delivered to a tumor seems to have a strong threshold role for efficacy. In the context of human delivery to larger tumor volumes can the numbers of bacteria needed be successfully delivered?

In summary, I think work demonstrates an interesting advance to the potential clinical applications of engineered bacteria, providing inducible STING agonist activity in a human-ready attenuated *E. coli* Nissle 1917 chassis. However, the novelty and advances need to be contextualized better by the authors through more extensive comparison to previous studies in the area and backed up by more rigorous data analysis and presentation before acceptance for publication.

Below Reviewer comments are listed in their entirety first and are then followed by a point-by-point reply to each individual comment.

Reviewer #1

In their manuscript, Leventhal et al. describe a novel constructed strain of probiotic *E. coli* Nissle 1917 for tumor therapy. This is very important work since novel immunotherapy strategies are required to fight this devastating disease. The authors introduce into the bacteria the ability to synthesize dicyclicAMP a second messenger mainly used by Gram-positive bacteria which is known to activate in mammalian cells intracellular signaling via the STING pathway. In addition, they introduce two attenuations of the bacteria with the aim that this strain will be usable in clinical trials. By extensive in vitro using murine and human lines as well as by in vivo testing in mice they are able to show the safety and efficacy of the bacteria against several transplantable tumors. In addition, they provide evidence that the treatment of tumors in mice elicits and CD8 killer T cell response responsible for the tumor clearance in some of the animals.

This is a very intensive and interesting study and needs to be presented to the general readership of Nature communications. Especially interesting is the fact that the authors restrict their treatment to intratumoral application. The bacteria were constructed to address the antigen presenting cells that activate the anti-tumor T cell response. Hardly any bacteria are found outside the tumors. This implies that the activation of such T cells takes place in the tumor itself and not in the draining lymph node. This could be a very important finding and should be discussed in the manuscript. However, before accepting this manuscript for publication that a few points need to be addressed. It presently feels rather incomplete.

1. The title is very unfortunate and does not provide information on the target of their synthetic biology attempt. Please specify.
2. Line 50: TME as abbreviation should be introduced in the introduction when the term tumor microenvironment is used the first time.
3. Line 70: An extensive study on probiotic *E. coli* including Nissle as tumor therapy was published and is not quoted here (Kocijancic et al. Oncotarget 2016).
4. Line 78: Histology and immunohistology would be appropriate here to learn about the localization of the bacteria in the tumor.
5. Line 122: is 8 days correct?
6. Line 137: The nitrate reductase promoter was described by Leschner et al. (Nucleic Acid Res. 2012) as tumor specific. This should be quoted.
7. Line 155: the authors claim that their bacteria containing a double attenuation should be unable to proliferate in tumors. They have not formally proofed that. In our hands Dasd still have a limited capacity to survive and proliferate in mice. A more cautious statement should be made here.
8. Line 164: The author claim to have used APC but really employed RAW264.7.
9. Lines 200 and following. Very important finding that phagocytosis is required for induction of IFN γ by SYN1891.
10. Line 225: PCR is used to detected SYN1891 in blood. It would be interesting whether they are still viable and can be determined as CFU (see comment 7).
11. Line 230: Several animals cleared B16 already after a single application of SYN1891. This is a very important finding. Very few attempts using bacteria were able to clear the B16 tumor under these conditions. Why were these animals not used for the tumor re-challenge? This is very important for generalizing the results.
12. Line 272: The authors claim a durable anti-response in multiple murine tumor while they nly showed it for two.
13. M&M and discussion: A very extensive study on direct application of therapeutic bacteria into the tumor has been published by Kocijancic et al. (Oncotarget 2017). The authors should discuss this and refer to it.
14. Line 342: the authors refer to serum sensitivity of ECN. Actually ECN is very resistant to human serum. However, the attenuations especially the deletion of *asd* might alter this dramatically. Therefore resistance of SYN1891 a gainst serum of different donor should be tested.

15. Discussion: Throughout their work, the authors use intratumor injection to apply their therapeutic bacteria. This is very much in accordance to the application of Coley's toxin because he usually treated skin tumors. The authors largely ignore the need to apply this therapy also to deeper tumors. This needs to be discussed.
16. Discussion: In their in vitro experiments, the authors show that phagocytosis is essential for SYNB1891 to elicit positive immune reactions. However, thus far it is clear from many studies that bacteria in the tumor are never found intracellular. This was also shown for different probiotic *E. coli* (Kocijacic et al. Oncotarget 2016). How do the authors reconcile these data with their findings on the requirement of phagocytosis.
17. Figure 3: The micrograph in Fig. 3e are of extremely low quality. Better quality pictures should be provided.
18. Figure 4f-g: TNFa, IFN γ and IL-1 β should be measured and shown.
19. Figure 5: Correct overlaps in Fig. 5a, c and d.
20. Supplementary Fig. 1a: Nothing can be recognized here. Provide better pictures.
21. Supplementary Fig. 6b-d: Provide data on TNFa and IL-1 β .

Reviewer #2:

In this manuscript, Leventhal and colleagues describe the construction and testing of an engineered bacterial strain, SYNB1891, which expresses cyclic-di-AMP (CDA) from an *E. coli* Nissle 1917 chassis. They demonstrate the progressive development of a CDA expression cassette along with various attenuations that make the strain compatible for future human delivery. The authors then test the final version of the strain, showing functionality as a STING agonist in in vitro cultures of mouse-derived cells, in mouse tumor models and in in vitro cultures of primary human DCs.

The concept of engineering bacteria as cancer therapeutics, including strains aiming to generate anti-tumor immunity, has been developed in several contexts over the past decades – including several clinical trials. Nevertheless, this work represents one of the more comprehensive examples of preclinical testing of synthetically engineered bacteria in the context of therapeutic delivery to date. The manuscript is broadly well written and well described (with a few exceptions mentioned below) and presents a strong case for further testing of CDA expressing *E. coli* to target tumors via the STING pathway. The work will likely be of interest in the context of immunotherapy and synthetic biology.

Although I am excited for the potential of this work, several aspects of the data analysis, presentation and interpretation need to be addressed before the work can be recommended for publication.

1 – Data analysis and experimental detail provided

Throughout the manuscript the level of detail provided regarding individual experiments and their data analysis is insufficient to properly judge the work. In particular:

- a. the actual number of replicates needs to be provided for each experiment/group
- b. where multiple groups are being compared there is currently no indication that the statistics provided take into account an appropriate multiple comparisons adjustment. The statistics must take this into account so if they currently do not the experiments should be re-analyzed, and the details of the tests used along with all comparisons made should be provided in each case.
- c. Levels of sensitivity should be indicated in relevant tests through the work. For example, the lack of detection of live bacteria or bacterial DNA in blood is only relevant in the context of the unstated sensitivity of the tests used.

d. With particular reference to the in vivo mouse tumor model experiments, the experimental designs were hard to follow. Readers would benefit considerably from schematics being added to the relevant figures to clearly demonstrate the numbers and groups treated alongside each experiment.

e. The imaging analysis in Supplementary Fig 1a needs more clarity on how it was analyzed and the implications of the outcomes.

2 – Overstatement of individual results.

a. The number of samples tested in many of the individual experiments/timepoints is relatively low. While the story as a whole is convincing based on its overlapping lines of evidence, in several cases the individual experiments undertaken have low n combined with high variation and thus the authors need to take care not to overstate individual results and their interpretations. Eg.

i) it is hard to conclude anything from the single image panels in Fig 3 d-f, particularly with no details regarding how a bacterium was determined as being “associated” with a cell or not (a simple contrast adjustment on the images suggests that some of the bacteria with no arrows could be associated with cells. Further, what do the arrows mean?). Either counts of multiple fields of view and quantitative information on the analysis criteria should be provided to support the authors claims that CytoD “significantly inhibited the number of bacterial cells observed” or the claims should be far more circumspect in the text.

ii) The data relating to late t-cell response in the tumor form Fig 2/Supp Fig 2, shows a couple of examples of high response in some EcN mice, even at day 2, which would seem to go against the authors’ hypothesis. The variability of this data should be reflected more clearly in the text.

b. in the absence of blinding of the treatment and control groups, the reliance on subjective measures for the total clearance of tumors (combined with no information provided in the methods other than “mice having no visible tumor on dissection”) makes these measures hard to interpret, particularly with low numbers of mice. Again, this should be reflected more clearly in the text.

3 – Lack of display of individual datapoints

With the number of samples used throughout the study, all datapoints should be displayed individually on graphs - in every case. This is currently not the case in several panels – (it appears that in fig 2b, for example, an n=2 dataset is being displayed as a mean +/- STD). Similarly, all tumor weight and volume graphs and some metabolite quantification graphs need to have individual datapoints displayed.

4 – Off target testing

Throughout the manuscript there are several references to off target effects of current STING agonists, with the suggestion that targeting through the use of EcN may overcome this issue. While the results of CytoD treatment to prevent phagocytosis by phagocytic cells are shown, the conclusions would be strengthened if the effect of EcN on off target cells was tested directly to substantiate these claims.

5 – Lack of sequence and resource sharing detail

While it is recognized that the authors have stated that “all DNA sequences and detailed schemes for genomic insertion and deletions used” are available upon request, there is no reason that these details should not be

included directly within the supplementary materials at publication (or full genome sequences deposited in an open-source sequence repository), as is routine practice for synthetic biology publications.

Similarly, there is no mention of the availability and sharing of the strains, which should be made available for example by deposit with addgene.

These aspects are critical to allow other researchers in the field to reproduce the work.

6 – Discussion of the work in the broader context of therapeutic bacterial delivery.

While there is a lot of attention given to the advances this approach may give over alternative available STING agonists, the authors need to more extensively cite and discuss previous examples of anti-tumor immunity generation using engineered bacteria and how this study fits within that context. In particular, several studies have utilized attenuated *Listeria monocytogenes*, which also target phagocytic cells and naturally secrete CDA to activate STING pathways (indeed the *dacA* gene used in this study is from *Listeria monocytogenes*), leading to the generation of anti-tumor immunity (eg. Gunn et al, *J Immunol* 2001; Singh et al, *J Immunol* 2005; Shahabi et al, *Immunother*, 2008). These warrant further discussion.

Additionally, in light of the previous lack of strong efficacy for bacterial anticancer therapeutics in the clinic to date the authors should discuss potential shortfalls of the approach. As one example, the amount of bacteria delivered to a tumor seems to have a strong threshold role for efficacy. In the context of human delivery to larger tumor volumes can the numbers of bacteria needed be successfully delivered?

In summary, I think work demonstrates an interesting advance to the potential clinical applications of engineered bacteria, providing inducible STING agonist activity in a human-ready attenuated *E. coli* Nissle 1917 chassis. However, the novelty and advances need to be contextualized better by the authors through more extensive comparison to previous studies in the area and backed up by more rigorous data analysis and presentation before acceptance for publication.

We thank the Reviewers for their thoughtful considerations and constructive feedback. We have used Reviewer comments to substantially improve the quality of our manuscript and generated additional data to further strengthen the conclusions drawn. Below are our point-by-point responses.

Original Reviewer comments appear in black

Our responses appear in blue with line numbers referring to the revised manuscript.

Changes to the main text are quoted and referenced in bold blue italics

Reviewer #1

In their manuscript, Leventhal et al. describe a novel constructed strain of probiotic E. coli Nissle 1917 for tumor therapy. This is very important work since novel immunotherapy strategies are required to fight this devastating disease. The authors introduce into the bacteria the ability to synthesize dicyclicAMP a second messenger mainly used by Gram-positive bacteria which is known to activate in mammalian cells intracellular signaling via the STING pathway. In addition, they introduce two attenuations of the bacteria with the aim that this strain will be usable in clinical trials. By extensive in vitro using murine and human lines as well as by in vivo testing in mice they are able to show the safety and efficacy of the bacteria against several transplantable tumors. In addition, they provide evidence that the treatment of tumors in mice elicits and CD8 killer T cell response responsible for the tumor clearance in some of the animals.

This is a very intensive and interesting study and needs to be presented to the general readership of Nature communications. Especially interesting is the fact that the authors restrict their treatment to intratumoral application. The bacteria were constructed to address the antigen presenting cells that activate the anti-tumor T cell response. Hardly any bacteria are found outside the tumors. This implies that the activation of such T cells takes place in the tumor itself and not in the draining lymph node. This could be a very important finding and should be discussed in the manuscript. However, before accepting this manuscript for publication that a few points need to be addressed. It presently feels rather incomplete.

1. The title is very unfortunate and does not provide information on the target of their synthetic biology attempt. Please specify.

We have modified the title of the manuscript to better clarity towards our approach.

Lines 1-2 “Immunotherapy with engineered bacteria: Targeting the STING pathway for anti-tumor Immunity”

2. Line 50: TME as abbreviation should be introduced in the introduction when the term tumor microenvironment is used the first time.

We thank the reviewer for noticing this typo. We have added the full description for the abbreviation TME when it is first mentioned in the text.

Lines 55-57 “In this study, we evaluated the utility of EcN as a platform for localized modulation of the tumor microenvironment (TME), demonstrating tumor-restricted colonization, intratumoral metabolic activity, localized immune activation and impact on tumor growth.”

3. Line 70: An extensive study on probiotic E. coli including Nissle as tumor therapy was published and is not quoted here (Kocijancic et al. *Oncotarget* 2016).

We appreciate the reviewer bringing this highly relevant reference to our attention. The reference has been added to the Introduction section

Lines 48-50 “Non-pathogenic bacteria with long historical use as probiotics and established safety profiles in humans, such as Escherichia coli Nissle 1917 (EcN) or Symbioflor-2, represent more favorable bacterial chassis for the development of a living cancer therapeutics^{17,18}.”

4. Line 78: Histology and immunohistology would be appropriate here to learn about the localization of the bacteria in the tumor.

While we can appreciate that analyzing intratumoral SYN1891 localization by histology or immunohistology would provide further characterization for the bacteria presented in this study, we do not believe it would substantially contribute to the primary findings presented. A variety of reports have already evaluated bacterial localization within various murine tumors models (S. Leschner, et al. *PLOS ONE*, 2009; T. Danino, et al. *ACS Synth. Biol.* 2012; M. Zhao, et al. *Proc. Natl. Acad. Sci.* 2005; V. Staedtke, et al. *Oncotarget* 2015; D. Kocijancic, et al. *Oncotarget* 2016). These studies suggest that intratumoral bacteria primarily reside within and around the necrotic core. Additionally, the development and optimization of such techniques within our group would not be trivial and take a substantial amount of time for manuscript resubmission. For these reasons, we believe that histological determination of intratumoral bacterial localization would be redundant with previous studies, would likely not provide novel insights or change the conclusions of the manuscript and would unnecessarily delay the publication of the study. We have updated the Introduction section to include brief remarks on the findings of these relevant studies.

Lines 44-47 “More recently, numerous studies have demonstrated the intrinsic capacity of various bacteria to selectively colonize tumors, primarily localizing to the hypoxic tumor core and sometimes leading to tumor regressions¹¹⁻¹⁷.”

5. Line 122: is 8 days correct?

We have clarified the experimental design referred to in line 122 of the main text and presented within Supplementary Figure 2a-c. As described in the figure legend, treatment was administered intratumorally on study days 1, 5 and 8. Tumor volumes were measured on study days 1, 5 and 8 (before injection). Cohorts of mice were euthanized on study days 2 and 9 for ex vivo analysis (24 hours and 8 days post dose initiation, respectively). Lines 129-130 note that T-cell associated cytokines are increased in tumors treated with the STING-agonist prototype strain at 8 days post dose initiation and is thus accurate. We have modified the main text for clarity.

Lines 126-130 “While treatment with both engineered and non-engineered EcN led to early increases in innate associated cytokines (like TNF α , GM-CSF, IL-6, IL-1 β and CCL2) 24 hours post dose initiation (Supplementary Fig. 2d), only SYN-Ptet-dacA treatment resulted in a shift to expression of T-cell associated cytokines (like IL-2, Granzyme B and IFN γ) at 8 days post dose initiation (Supplementary Fig. 2e).”

6. Line 137: The nitrate reductase promoter was described by Leschner et al. (*Nucleic Acid Res.* 2012) as tumor specific. This should be quoted.

We thank the reviewer for the suggested reference. The data presented in this study supports the conclusion that a hypoxia inducible promoter can be utilized to enrich expression of therapeutic payloads within the hypoxic tumor microenvironment relative to anoxic healthy tissue. We have modified the main text to include this information.

Lines 145-150 “While all promoters showed functionality within an hour of administration of their respective inducing agent, the hypoxia inducible P_{fnrS} ^{34,35} was selected for further development as this promoter showed the highest levels of intratumoral payload induction, with 65-95% of all bacteria induced. These data are in agreement with the results reported for *Salmonella typhimurium* where a set of hypoxia sensitive promoters, such as the Nitrate Reductase promoter, were shown to enable tumor-specific induction compared to the spleen³⁶.”

7. Line 155: the authors claim that their bacteria containing a double attenuation should be unable to proliferate in tumors. They have not formally proofed that. In our hands Dasd still have a limited capacity to survive and proliferate in mice. A more cautious statement should be made here.

We appreciate the reviewer's attention to detail in reference to the formal conclusions drawn by the data shown in Figures 2F, 4A and Supplementary Figures 2f,g, 4b,c. Unlike WT EcN, or EcN with a thymidine auxotrophy, which expand and persist in tumors long-term, EcN strains harboring a diaminopimelic acid auxotrophy fail to expand in number compared to the quantity of cells injected and are cleared from tumors over time. While it is possible that these cells are capable of proliferating as individuals, as a group they fail to persist, their intratumoral concentrations reduce and are eventually undetectable by colony forming unit assay at later time points. We appreciate the reviewers concern with our wording and have modified the main text accordingly.

Lines 161-166 “Diaminopimelic acid (dap) is a component of the bacterial cell wall and is not produced by eukaryotes, and therefore we hypothesized that a dap auxotrophic strain would be unable to survive in a mammalian host environment. Indeed, deletion of the *dapA* gene (encoding 4-hydroxy-tetrahydropicolinate synthase) resulted in a mutant strain of EcN that was unable to expand or persist within tumors and was cleared over time (Supplementary Fig. 2f).”

8. Line 164: The author claim to have used APC but really employed RAW264.7.

The main text has been modified accordingly.

Lines 173-177 “The final clinical candidate strain, referred to as SYN1891, maintained its anaerobically inducible production of CDA (Fig. 2g), dose-dependent biological activity when co-cultured with RAW 264.7 macrophage cells in vitro (Fig. 2h), inherent sensitivity to human serum (Supplementary Fig. 2h) and sensitivity to a wide panel of antibiotics currently utilized in the clinic (Supplementary Fig. 2i).”

9. Lines 200 and following. Very important finding that phagocytosis is required for induction of IFN β by SYN1891.

We agree with the reviewers comment and appreciate their enthusiasm for this finding. Since we utilize a bacterium to produce, package and deliver the STING-agonist payload, it was important to evaluate the role of phagocytosis in SYN1891's mechanism of action. Considering that EcN does not actively invade through the mammalian cell membrane and is not able to passively enter the cytoplasm we hypothesized that STING activation would be phagocytosis dependent. We compared SYN1891 to a small-molecule STING agonist which can be internalized by many cell types via pinocytosis. The findings presented in Figure 3 and Supplementary Figure 3 support the conclusion that phagocytosis is required for SYN1891 to enter antigen presenting cells and subsequently induce Type I interferon responses, and thus suggests that SYN1891 activity would be enriched within or restricted to phagocytic cells inside the tumor.

10. Line 225: PCR is used to detected SYN1891 in blood. It would be interesting whether they are still viable and can be determined as CFU (see comment 7).

The presence and viability of WT EcN and SYN1891, which contains both a Dap and Thy auxotrophy, were assessed by colony forming unit (CFU) assay of whole blood at various dose levels and at various

timepoints following injection into tumor bearing mice. Data presented in Figure 4a and Supplementary Figures 4a, 4b show a complete lack of detectable SYN1891 by CFU assay of whole blood taken at 1, 3, 7 and 10 days post intratumoral injection (regardless of dose level). Additionally, data presented in Figure 1c show no detectable CFUs for WT EcN in whole blood at 1.5 hours, 24 hours and 8 days post intratumoral injection. These data would suggest that even if bacterial cells were present in the blood at any of these time points, they are likely not viable.

11. Line 230: Several animals cleared B16 already after a single application of SYN1891. This is a very important finding. Very few attempts using bacteria were able to clear the B16 tumor under these conditions. Why were these animals not used for the tumor re-challenge? This is very important for generalizing the results.

We appreciate the reviewer's enthusiasm and we shared an equal level of excitement for the robust responses we observed with the SYN1891 treatment on a murine model as aggressive as the B16F10 melanoma model. While we observed significant delays in tumor growth for this model (Figures 4b and 5a), the complete response rates remained low relative to the A20 tumor model (30% on average for B16F10 vs. up to 80% for A20). With a 30% complete response rate we would require 35+ mice per group to produce a large enough pool of tumor-free mice for re-challenge studies. With our ultimate goal of generating data to support the transition into clinical studies, we thus decided to perform re-challenge studies in the A20 model.

12. Line 272: The authors claim a durable anti-response in multiple murine tumor while they only showed it for two.

The main text has been modified accordingly to accurately represent the formal conclusions that can be drawn from the data presented.

Lines 289-292 "Collectively, these data demonstrate the complementary contributions of both the EcN bacterial chassis and the production of the STING agonist CDA towards the generation of an efficacious and durable antitumor immune response in two distinct murine tumor models (B16F10 and A20) in two genetic backgrounds (C57BL/6 and BALB/c)."

13. M&M and discussion: A very extensive study on direct application of therapeutic bacteria into the tumor has been published by Kocijancic et al. (Oncotarget 2017). The authors should discuss this and refer to it.

We thank the reviewer for bringing this reference to our attention. To better contextualize the work conducted in this paper we have expanded discussions of various bacterial based approaches in the Discussion section and added this reference as well as reference to several other relevant studies. Please refer to lines 379-408 in the main text and the reply to Reviewer #2, comment 6, below.

14. Line 342: the authors refer to serum sensitivity of ECN. Actually ECN is very resistant to human serum. However, the attenuations especially the deletion of *asd* might alter this dramatically. Therefore resistance of SYN1891 against serum of different donor should be tested.

We respectfully disagree with the reviewers comment that EcN is resistant to human serum. Historically EcN has been known for its serum sensitivity, which has been primarily attributed to its semi-rough, truncated lipopolysaccharide (LPS) layer. This was formally demonstrated by Grozdanov L. et al. in 2002 where they identified a single point mutation in the O-antigen polymerase gene *wzy* of EcN which results in the production a truncated O6 LPS and semirough phenotype (Grozdanov L et al. *J. Bacteriol.* 2002). Insertion of a functional copy of the *wzy* gene into EcN results in elongation of the O6 LPS sidechain and imparts resistance to human serum. In the absence of a functional *wzy* gene (e.g. WT EcN) exposure to either 50% or 90% human serum resulted in zero viability after 1 hour of treatment. In contrast, while it never rendered complete resistance to human serum, EcN given a functional *wzy* gene had detectable viability up to 24 hours post-exposure.

Nevertheless, to formally confirm serum sensitivity we performed additional testing where SYNB1891 was exposed to either normal or heat inactivated serum from three healthy human donors and the impact on viability was determined by a colony forming unit assay (Supplementary Figure 2h). While viable bacteria were detected following up to a 180-minute exposure to heat inactivated serum, no viable bacteria were detected after 45 minutes of exposure to non-inactivated serum. The main text, methods and supplementary figures have been updated to include these results.

Lines 173-177 “The final clinical candidate strain, referred to as SYNB1891, maintained its anaerobically inducible production of CDA (Fig. 2g), dose-dependent biological activity when co-cultured with RAW 264.7 macrophage cells in vitro (Fig. 2h), inherent sensitivity to human serum (Supplementary Fig. 2h) and sensitivity to a wide panel of antibiotics currently utilized in the clinic (Supplementary Fig. 2i).”

Lines 833-841 “Human serum (complement) sensitivity. Fresh healthy human serum samples from 3 donors were obtained from Research blood components, Watertown, MA. Aliquots of serum from each donor were heat inactivated at 56°C for 30 min (after aliquots reached the temperature). Induced SYNB1891-cmR bacteria (see Anerobic in vitro circuit induction) were resuspended at a concentration 1×10^7 cells/ml and 50 μ l of bacteria were added to 450 μ l of human serum (untreated or heat-inactivated). Next, bacteria were incubated at 37°C with shaking for T = 0, 45, 90 and 180 minutes and then plated in serial dilutions for CFU counting on LB-agarose plates supplemented with thymidine, diaminopimelic acid and chloramphenicol (see Bacterial quantification from in vivo samples) to determine susceptibility to serum lysis.”

15. Discussion: Throughout their work, the authors use intratumor injection to apply their therapeutic bacteria. This is very much in accordance to the application of Coley’s toxin because he usually treated skin tumors. The authors largely ignore the need to apply this therapy also to deeper tumors. This needs to be discussed.

This is an important aspect of the clinical evaluation of SYNB1891. We have modified the Discussion section accordingly to briefly mention this consideration.

Lines 401-403 “While safety and tolerability will first be assessed for the percutaneous route of administration, future investigations with radiologically guided intratumoral injection would enable treatment of visceral lesions.”

16. Discussion: In their in vitro experiments, the authors show that phagocytosis is essential for SYNB1891 to elicit positive immune reactions. However, thus far it is clear from many studies that bacteria in the tumor are never found intracellular. This was also shown for different probiotic E. coli (Kocijacic et al. Oncotarget 2016). How do the authors reconcile these data with their findings on the requirement of phagocytosis.

We agree with the reviewer’s conclusions that non-pathogenic bacterial species, like EcN, would predominantly exist in the extracellular space of the tumor microenvironment as they lack the virulence factors and cell membrane penetrating machinery to actively invade mammalian cells. This is in direct contrast to pathogenic vectors previously utilized, such as *Salmonella* or *Listeria*, which can be found intracellularly invading various mammalian cell subsets. Techniques previously utilized to evaluate bacterial localization in tumors typically involved histology, immunofluorescence or electron microscopy, all of which only provide a relatively limited window (both spatially and temporally) to ascertain the localization of bacterial cells throughout the tumor. While the authors of the referenced study using probiotic Symbioflor-2 (D. Kocijacic et al. *Oncotarget* 2016) did not find intracellular probiotic E. coli by Electron Microscopy, after evaluating phagocytosis and intracellular survival of these bacteria using *in vitro* assays they still conclude that probiotic E. coli “should be considered controllable by macrophage uptake *in vivo*”. Since phagocytosed (yet intact) bacteria within dendritic cells or macrophages are likely a relatively rare event at any moment in time compared to all bacteria inside the tumor, it is difficult to make definitive conclusions from a few micron sized samplings of tumor tissue. Indeed, techniques capable of analyzing larger proportions of the intratumoral

space would need to be developed to trace phagocytosis events within the tumor and more accurately ascertain the frequency of such events *in vivo*. The development of such techniques is beyond the scope of this manuscript. However, *in vitro* data presented in Figure 3 demonstrate that active phagocytosis is critical for the entry of SYN1891 in dendritic cells and subsequent stimulation and release of type I interferons. In Figure 4, following intratumoral injection *in vivo*, SYN1891 treatment leads to significant upregulation of type I interferons. While phagocytosed SYN1891 may be relatively rare *in vivo* compared to extracellular SYN1891, these combined data would suggest that such events do occur and are biologically significant toward the development of anti-tumor immunity.

17. Figure 3: The micrograph in Fig. 3e are of extremely low quality. Better quality pictures should be provided.

We apologize if any of the figures appear low quality and ensure that high resolution images will be submitted in the final format of the manuscript.

18. Figure 4f-g: TNF α , IFN γ and IL-1 β should be measured and shown.

We thank the reviewer for pointing out this omission. For brevity, measurements of these three cytokines were not included in the initial submission. However, for consistency and to provide a more complete profiling of intratumoral cytokine response to SYN1891, Figure 4 now includes readouts for TNF α (Figure 4f), IFN γ (Figure 4i) and IL-1 β (Figure 4h).

19. Figure 5: Correct overlaps in Fig. 5a, c and d.

We are unaware of any formatting errors in Figure 5, if the reviewer could provide further clarity or if the copy editor notes such issues, we will happily correct them for the final formatting of the manuscript.

20. Supplementary Fig. 1a: Nothing can be recognized here. Provide better pictures.

Please see response to comment #17.

21. Supplementary Fig. 6b-d: Provide data on TNF α and IL-1 β .

We again thank the reviewer for bringing these readouts to our attention. For brevity, these measurements were excluded from the initial submission as they exhibited similar patterns of expression to IL-6. For consistency and completeness, they have now been included in Supplementary Figure 6f-g (TNF α) and Supplementary Figure 6h-i (IL-1 β) for both timepoints analyzed.

Reviewer #2:

In this manuscript, Leventhal and colleagues describe the construction and testing of an engineered bacterial strain, SYN1891, which expresses cyclic-di-AMP (CDA) from an *E. coli* Nissle 1917 chassis. They demonstrate the progressive development of a CDA expression cassette along with various attenuations that make the strain compatible for future human delivery. The authors then test the final version of the strain, showing functionality as a STING agonist in *in vitro* cultures of mouse-derived cells, in mouse tumor models and in *in vitro* cultures of primary human DCs.

The concept of engineering bacteria as cancer therapeutics, including strains aiming to generate anti-tumor immunity, has been developed in several contexts over the past decades – including several clinical trials. Nevertheless, this work represents one of the more comprehensive examples of preclinical testing of synthetically engineered bacteria in the context of therapeutic delivery to date. The manuscript is broadly well written and well described (with a few exceptions mentioned below) and presents a strong case for further testing of CDA expressing *E. coli* to target tumors via the STING pathway. The work will likely be of interest in the context of immunotherapy and synthetic biology.

Although I am excited for the potential of this work, several aspects of the data analysis, presentation and interpretation need to be addressed before the work can be recommended for publication.

1 – Data analysis and experimental detail provided

Throughout the manuscript the level of detail provided regarding individual experiments and their data analysis is insufficient to properly judge the work. In particular:

a. the actual number of replicates needs to be provided for each experiment/group

We agree with the reviewer's recommendation and have made the appropriate changes to the data displayed to improve clarity. The following adjustments have been made to all figures and legends:

- When feasible, the number of replicates is displayed as a single dot representing an individual biological replicate. Additionally, each dot is now outlined to improve contrast and better resolve individual data points.
- When group means are displayed in the main figures the number of replicates per group are referenced in the figure legend. For clarity, we have also added a corresponding plot in the supplementary figures which displays all individual data points. For example, in Figure 1, panels a-e have individual dots for each independent biological replicate while panel f shows the mean tumor volume for each experimental group. The total number of mice per group is noted for Figure 1f in the figure legend. Additionally, in Supplemental Figure 1d all individual measurements are displayed for all groups and time points for reference and comparison. Similar accommodations have been made for Figure 4b and all efficacy studies displayed in Figure 5.

b. where multiple groups are being compared there is currently no indication that the statistics provided take into account an appropriate multiple comparisons adjustment. The statistics must take this into account so if they currently do not the experiments should be re-analyzed, and the details of the tests used along with all comparisons made should be provided in each case.

We sincerely appreciate the reviewer identifying this important experimental detail. Upon further consideration we have reanalyzed all relevant datasets throughout the manuscript utilizing one-way or two-way ANOVA with Tukey's multiple comparisons posttests where appropriate. While these new analyses did not alter the fundamental conclusions originally drawn, they substantially improve the quality of the manuscript by providing a more relevant and more stringent statistical analysis for comparing the results between various experimental groups. All figures, figure legends and methods have been updated accordingly. When feasible, for clarity we have also added exact P values within the figure legends. Below is the addition made to the Methods section to clarify these changes.

Lines 947-957 "Data analysis. Prism7.0 c software (GraphPad Software, San Diego, CA) was used for all statistical analysis. The details of the statistical tests carried out are indicated in the respective figure legends. Values were compared using either a Student's t-test to compare two groups or one-way ANOVA with Tukey's multiple comparisons posttests to compare the variance in the multiple group means with one independent factor (e.g. treatment group). We employed two-way ANOVA with Tukey's multiple comparisons posttests when we had the effect of two factors (e.g. treatment and time (or genotype)) on a dependent variable. In in vivo experimental cases where some mice were removed from study for a treatment group upon reaching a study endpoint, we performed comparisons between the groups using one-way ANOVA with Tukey's multiple comparisons posttests at the indicated timepoint. For Kaplan-Meier survival experiments, we performed a log-rank (Mantel-Cox) test."

c. Levels of sensitivity should be indicated in relevant tests through the work. For example, the lack of detection of live bacteria or bacterial DNA in blood is only relevant in the context of the unstated sensitivity of the tests used.

We appreciate the reviewer's comments related to the sensitivity of detection for both bacterial quantification by Colony Forming Unit (CFU) assay and qPCR detection of engineered bacterial genomic DNA. These are important parameters to consider and we have updated the methods section accordingly.

Lines 765-767 "CFU bacterial quantification exhibited a limit of detection of ~100 viable bacterial cells per milliliter of sample with a limit of quantification of ~3000 bacterial cells per milliliter."

Lines 773-782 "For the detection of SYN1891 bacterial DNA in blood and tumor a SYN1891 probe specific qPCR assay was developed and validated using good laboratory practices standards at QPS, LLC, Newark DE, USA. Tumors and EDTA treated blood were snap frozen and shipped to QPS for analysis. The qPCR analysis was performed using customized SYN1891 forward, reverse primers and probe (Life Technologies, ThermoFisher Scientific) according to analytical procedure (method) number 1813-1804. SYN1891 primer/probe specific sequences were as follows: Forward primer "5'- AGGATACTCATGTTGGAAAAGTCCAT-3'"; Reverse primer "5'- CCCGCATCCCACGCTAAC-3'"; Probe "5'-6FAM CTTGCCCCCTTATATMGBNFQ-3'". The qPCR detection limit was 10 copies of SYN1891 DNA per 1 µg of total DNA with a quantification limit of 25 copies per sample."

d. With particular reference to the in vivo mouse tumor model experiments, the experimental designs were hard to follow. Readers would benefit considerably from schematics being added to the relevant figures to clearly demonstrate the numbers and groups treated alongside each experiment.

Experimental outlines have been added for all *in vivo* studies. These outlines include the tumor model utilized, the treatments administered, dosing frequency/timing and, in some cases, when cohorts of mice were analyzed for the indicated endpoint. Treatment dose level are indicated in panel legends and the number of mice treated per group are noted in figure legends. We hope this will provide better clarity and easier reference for the reader.

e. The imaging analysis in Supplementary Fig 1a needs more clarity on how it was analyzed and the implications of the outcomes.

The bioluminescence imaging displayed in Supplementary Figure 1a shows the macroscopic localization of the engineered EcN strain within and outside of the B16F10 melanoma tumor at 30 min, 24 hours and 72 hours post intratumoral injection. These data are intended to be qualitative, illustrating the macroscopic localization of the bacteria versus the surrounding tissue. The main text has been modified to further clarify the qualitative nature of these data. The methods utilized to capture these images are described in the "Methods" section in the subsection labeled "Bacterial quantification from *in vivo* samples and bioluminescence detection" and have been updated for clarity as well. As described in the figure legend, the white light exposure images were utilized to delineate the border between the tumor mass and surrounding subcutaneous tissue. This border was then overlaid onto the bioluminescence image to illustrate the localization of the bacteria. If the images appeared blurry, this issue will be resolved with submission of high-resolution images for the final version of the manuscript.

Lines 83-87 "Bioluminescence also enabled qualitative monitoring of bacterial localization within the tumor and surrounding tissue on a macroscopic scale. Following i.t. injection bioluminescence is detected within and throughout the tumor mass, however it is not detected in the subcutaneous space surrounding the tumor up to 72 hours post injection (Supplementary Fig. 1a)."

Lines 769-773 "To detect bacterial luminescence for excised tumor and/or surrounding tissue, samples were placed in a G:Box (Syngene) gel doc system. Images under white light were captured to delineate the border between the B16F10 tumors and the surround healthy subcutaneous skin tissue. Bacterial luminescence was capture in the absence of light with full aperture and a 2 min exposure time."

2 – Overstatement of individual results.

a. The number of samples tested in many of the individual experiments/timepoints is relatively low. While the story as a whole is convincing based on its overlapping lines of evidence, in several cases the individual experiments undertaken have low n combined with high variation and thus the authors need to take care not to overstate individual results and their interpretations. Eg.

i) it is hard to conclude anything from the single image panels in Fig 3 d-f, particularly with no details regarding how a bacterium was determined as being “associated” with a cell or not (a simple contrast adjustment on the images suggests that some of the bacteria with no arrows could be associated with cells. Further, what do the arrows mean?). Either counts of multiple fields of view and quantitative information on the analysis criteria should be provided to support the authors claims that CytoD “significantly inhibited the number of bacterial cells observed” or the claims should be far more circumspect in the text.

We agree with the reviewer’s comments that additional analyses are needed to formally reach the conclusions stated and clarifications need to be made for how phagocytosis is measured via microscopy in the assays described in Figure 3. Figures 3d-f are representative fields of view (FOV) demonstrating the observed close association of bacteria with dendritic cells *in vitro* and colocalization of those bacteria with the actin cytoskeleton and within Lamp-1 associated phagosomes. White arrows are intended to point out examples of these associations (or lack thereof for cytoD treated cells). The purpose of the white arrows has been clarified in the figure legend.

Lines 494-495 “White arrows point to bacteria co-localized within DC actin cytoskeleton (d-f). These bacteria are surrounded by phagosomal membrane stained with LAMP-1 (e).”

Additionally, we have performed new analyses whereby at least 10 FOVs (and >200 cells per group) were analyzed to quantify the average number of bacteria associated with a dendritic cell for each treatment group (with or without cyto D to inhibit phagocytosis). F-actin and Hoechst (nuclei) staining were utilized to identify the cellular boundaries of individual DCs, then anti-GFP staining was utilized to identify bacteria colocalized within the boundary of a DC. The average number of bacteria per DC in an FOV was then quantified and shown as the ratio of bacteria internalized by DCs to the total number of DCs per FOV. Utilizing this quantification method, we observed a highly significant 92% reduction of bacteria found associated with DCs when cells were treated with cyto D compared to non-treated controls (Figure 3g). The main text, main figures, supplementary figures and the methods section has been updated accordingly (see below). We believe these new data more clearly illustrate the impact of active phagocytosis on bacterial internalization. Additionally, when combined with functional data showing significant inhibition of *Ifnb1* expression following cytoD treatment (Figure 3h and 3j), these complimentary data strongly suggest that phagocytosis plays a critical role in the STING-agonistic mechanism of action of SYN1891.

Lines 210-215 “To quantify phagocytosed bacteria, nuclei and F-actin staining were used to identify individual BMDCs and demarcate their cellular border, respectively. Co-culture of murine BMDCs with SYN1891 modified to express GFP (SYN1891-gfp), showed many bacterial cells were internalized (e.g. co-localized with F-actin staining) and residing within mature phagosomes that contained lysosome-associated membrane protein (LAMP-1) (Fig. 3d,e).”

Lines 496-498 “The number of bacteria per dendritic cell per field of view (FOV) was quantified in (g). At least 10 FOVs were evaluated per experiment resulting in > 200 cells per treatment quantified (**P<0.0001, two-tailed unpaired Student’s t-test comparing control vs CytoD treated groups). Images and data are representative of 2 independent experiments.”

Lines 877-880 “A representative example for the quantification of bacteria colocalized within dendritic cells is shown in Supplementary Figure 10. The average number of bacteria per

dendritic cell in a field of view (FOV) was quantified and shown as the ratio of bacteria internalized by dendritic cells to the total number of dendritic cells per FOV.”

ii) The data relating to late t-cell response in the tumor form Fig 2/Supp Fig 2, shows a couple of examples of high response in some EcN mice, even at day 2, which would seem to go against the authors' hypothesis. The variability of this data should be reflected more clearly in the text.

We respectfully disagree with the reviewer's conclusion that data shown in Supplemental Figure 2d go against the central hypothesis and conclusions of the study that engineering the STING-activating circuit into EcN is critical for its ability to induce durable and efficacious antitumor immunity. However, we do appreciate the reviewer's observation that indeed EcN alone is capable of driving intratumoral inflammation which does ultimately slow tumor growth. While we note the moderate level of efficacy observed following treatment of murine tumors with control EcN, Figures 1f, 5a and 5b, we also note that this efficacy is limited to slowing of tumor growth and is not capable of driving durable rejection and tumor clearance. Cytokine expression profiling from both *in vitro* and *in vivo* analyses throughout the paper support these findings in that both wild type EcN and SYN1891 treatments result in upregulation of innate response associated cytokines (like IL-6, IL-12, TNF α , GM-CSF, IL-1 β and CCL2), however only SYN1891 is capable of driving robust IFN β expression (Figure 2c). As referenced in the reviewer's comment, a minority of control EcN treated tumors do exhibit upregulation of IL-2 and IFN γ (Supplemental Figure 2d), however only STING-agonist producing bacteria are capable of driving consistent upregulation of IL-2, Granzyme B and IFN γ . Taken in total these data points would suggest that the addition of a STING-agonistic, type I interferon inducing functionality combined with the broadly immune activating capacity of the EcN chassis provides the right context to drive efficacious antitumor immune responses. We appreciate that these are not direct functional readouts of T cell-based antitumor immunity (like those shown by the depletion of CD4 or CD8 T cells in Figure 5d), thus we have appropriately utilized words such as "suggest" and "possibly" (lines 129-131 in the main text) to describe the results depicted in the referenced Supplemental Figure 2.

b. in the absence of blinding of the treatment and control groups, the reliance on subjective measures for the total clearance of tumors (combined with no information provided in the methods other than "mice having no visible tumor on dissection") makes these measures hard to interpret, particularly with low numbers of mice. Again, this should be reflected more clearly in the text.

In experiments where terminal measurements were required, such as tumor weights and measurement of intratumoral bacterial abundance and cytokine levels (for example Figure 4b), complete responders (C.R.) were noted to be "mice having no visible tumor upon dissection" as these mice had no observable tumor mass to be dissected and measured. However, since complete response would imply total clearance of tumor cells and a lack of relapse or tumor reemergence long-term, we have modified the Figure to replace "C.R." with "No Detectable Tumor" and adjusted the Figure legend accordingly as shown below.

Lines 520-521 "Total tumor weight for treated mice at the indicated time points with mice having no visible tumor mass upon dissection (no tumor detected (N.T.D)) on Study day 10"

For efficacy experiments not requiring *ex vivo* terminal measurements, such as those depicted in Figure 5 and Supplemental Figure 5, the complete responder rate is shown for the final timepoint in which tumor volumes were actively measured. All mice having no palpable tumor at this timepoint were monitored until at least study day 60, often 90+ days, and at least 50 days following the final treatment dose. Mice were only considered complete responders if they exhibited no relapse or emergence of a palpable tumor long-term. We have updated the Methods section to accurately reflect this criterion.

Lines 737-739 "All complete responder (C.R.) mice, or those having no palpable tumor by the indicated timepoint, were monitored until at least study day 60 and at least 50 days post the final treatment dose to ensure no tumor relapse was observed."

3 – Lack of display of individual datapoints

With the number of samples used throughout the study, all datapoints should be displayed individually on graphs - in every case. This is currently not the case in several panels – (it appears that in fig 2b, for example, an n=2 dataset is being displayed as a mean +/- STD). Similarly, all tumor weight and volume graphs and some metabolite quantification graphs need to have individual datapoints displayed.

Please refer to comment 1a above.

4 – Off target testing

Throughout the manuscript there are several references to off target effects of current STING agonists, with the suggestion that targeting through the use of EcN may overcome this issue. While the results of CytoD treatment to prevent phagocytosis by phagocytic cells are shown, the conclusions would be strengthened if the effect of EcN on off target cells was tested directly to substantiate these claims.

In the Introduction we provide general context and background for the reader by referencing recent preclinical and clinical observations that “off-target” effects of untargeted, small molecule STING agonists may negatively impact the efficacy of such treatments. In the Discussion section, we reiterate the conclusions drawn from our *in vitro* studies that phagocytosis plays a critical role in the mechanism of action of SYN1891 and that this “is potentially an important feature to avoid “off-target” STING-pathway activation”. In these two instances, we have intentionally chosen words which indicate a purely speculative tone, like “potentially”, “may” and “potential” and have avoided making any conclusions within the main text which would suggest a definitive conclusion to this particular question. While we agree that direct evidence demonstrating a lack of “off-target” activation of STING in other cell types would provide value, the development of bacterial-T cell cocultures would represent unique technical challenges and would not be trivial to develop in a form that would provide meaningful answers to this question. Since this effort, if feasible, would result in a substantial delay in resubmission and since the central conclusions of this manuscript do not principally involve the definitive comparison of our approach versus small molecule STING agonists, we believe such studies would not substantially change the conclusions drawn.

5 – Lack of sequence and resource sharing detail

While it is recognized that the authors have stated that “all DNA sequences and detailed schemes for genomic insertion and deletions used” are available upon request, there is no reason that these details should not be included directly within the supplementary materials at publication (or full genome sequences deposited in an open-source sequence repository), as is routine practice for synthetic biology publications.

Similarly, there is no mention of the availability and sharing of the strains, which should be made available for example by deposit with addgene.

These aspects are critical to allow other researchers in the field to reproduce the work.

As members of the synthetic biology community we appreciate the reviewers concerns and have added all relevant sequences to the Methods and Supplementary Data sections. Previously we had created a full list of all strains utilized throughout the manuscript with detailed information on any modifications made (see Table 1). The full genome sequence of *E. Coli* Nissle 1917 is already publicly available through the Genbank/EMBL/DDBJ (accession number: CP007799). We have added sequences and plasmid maps for the CDA producing enzymes evaluated (Supplementary Figure 8 and Supplementary Table 1) and the genomic manipulations made in the final SYN1891 clinical candidate strain (Supplementary Figure 9 and Supplementary Tables 2-5). All engineered strains will be made available subject to an MTA. For clarity the following was added to the Methods section.

Lines 678-685 “Complete genome sequence of E.coli Nissle 1917 is listed in Genbank/EMBL/DDBJ accession number: CP007799. Plasmid maps and DNA sequences for CDA producing enzymes are listed in the Supplementary Table 1 and Supplementary Figure 8. The genome map and manipulated genome sequence of SYN1891 is available in Supplementary Figure 9 and Supplementary Tables 2-5. Engineered strains described in this manuscript can be made available subject to a Material Transfer Agreement (MTA), which can be requested by contacting the corresponding authors. All requests will be reviewed by Synlogic to verify whether the request is subject to any intellectual property or confidentiality obligations.”

6 – Discussion of the work in the broader context of therapeutic bacterial delivery.

While there is a lot of attention given to the advances this approach may give over alternative available STING agonists, the authors need to more extensively cite and discuss previous examples of anti-tumor immunity generation using engineered bacteria and how this study fits within that context. In particular, several studies have utilized attenuated *Listeria monocytogenes*, which also target phagocytic cells and naturally secrete CDA to activate STING pathways (indeed the *dacA* gene used in this study is from *Listeria monocytogenes*), leading to the generation of anti-tumor immunity (eg. Gunn et al, *J Immunol* 2001; Singh et al, *J Immunol* 2005; Shahabi et al, *Immunother*, 2008). These warrant further discussion.

Additionally, in light of the previous lack of strong efficacy for bacterial anticancer therapeutics in the clinic to date the authors should discuss potential shortfalls of the approach. As one example, the amount of bacteria delivered to a tumor seems to have a strong threshold role for efficacy. In the context of human delivery to larger tumor volumes can the numbers of bacteria needed be successfully delivered?

We thank the reviewer for bringing forth these considerations for discussion. We agree that a brief review comparing previous bacterial based approaches with SYN1891 would be beneficial to improve the quality of the manuscript and better contextualize the novelty and advances made. A discussion of considerations related to its clinical evaluation would also be helpful. To this end, we have supplemented the Discussion section to review these topics.

*Lines 379-408 “A variety of bacterial based therapies have previously been investigated for the treatment of cancer, with the majority using attenuated pathogens such as *Salmonella typhimurium*, *Clostridium novyi*, and *Listeria monocytogenes*¹¹. *Listeria* based approaches typically use attenuated strains as a vaccine vector, taking advantage of the intrinsic capacity of the bacterium to forcefully invade APCs and engineering expression of shared tumor-associated antigens or unique neoepitopes personalized to each patient⁴⁴⁻⁴⁷. Similar to traditional vaccine approaches, this requires the a priori identification of efficacious target antigens. While *Salmonella* and *Clostridium* based therapeutics also require the use of highly attenuated strains, such approaches primarily take advantage of the oncolytic potential of these pathogenic bacteria⁴⁸. Clinically these oncolytic strains have either failed to exhibit robust efficacy or have been hindered by dose-limiting toxicities¹¹. While alternative routes of administration, like intratumoral injection, may help to reduce toxicities for *Salmonella* based approaches⁴⁹, attempts to further improve efficacy by engineering in additional functions are often limited due to a lack of tools and methods for genetic manipulation. In contrast, SYN1891 uses the non-pathogenic strain *EcN*, which has a potentially advantageous safety profile and significantly broader set of tools for genetic manipulation and engineering. Using synthetic biology techniques we rationally designed an *EcN* strain that expresses the STING-agonist CDA and, following intratumoral injection, delivers targeted activation to APCs in the TME in order to generate and support antitumor immune responses. Since previous bacterial based therapies with compelling preclinical evidence have failed to demonstrate efficacy in patients, clinical testing of SYN1891 is critical to assess the translatability of our findings to human disease. A Phase I clinical trial of SYN1891 intratumoral injection in patients with percutaneous*

accessible advanced or metastatic malignancies has already been initiated (ClinicalTrials.gov Identifier: NCT04167137). While safety and tolerability will first be assessed for the percutaneous route of administration, future investigations with radiologically guided intratumoral injection would enable treatment of visceral lesions. Positive indications for safety and efficacy would provide support for the development of additional engineered strains with rationally designed functionalities tailored to the needs of specific cancer and patient subtypes. Although the application of synthetic biology to human therapeutics is still in its infancy, our work highlights its potential and provides guidance for the development of future approaches intended for clinical evaluation.”

In summary, I think work demonstrates an interesting advance to the potential clinical applications of engineered bacteria, providing inducible STING agonist activity in a human-ready attenuated E. coli Nissle 1917 chassis. However, the novelty and advances need to be contextualized better by the authors through more extensive comparison to previous studies in the area and backed up by more rigorous data analysis and presentation before acceptance for publication.

Reviewers' Comments:

Reviewer #1:

Remarks to the Author:

In the revised version of their manuscript Leventhal et al. have appropriately taken care of my concerns. A few minor points need still to be clarified to my mind.

1. In the second paragraph of my original review, I state that their data suggest that the anti-tumor response is induced directly in the neoplasm because they do not find any bacteria outside of the tumor. Presently, it is not known where the anti-tumor reaction is activated. This is important for future work on this type of therapy using bacteria carrying immune modulatory molecules or antigens. Therefore, I requested that this needs to be discussed although I did not state it in the bullet pointed review. Please discuss as I find this is a very important finding.
2. Phagocytosis: I disagree with the authors although at the moment I cannot prove them wrong. However, my point directs are something different. Bacteria own secretion systems which might target the cytosol of APCs without phagocytosis of the entire bacteria. In vitro they clearly demonstrated bacterial phagocytosis and by using cytochalasin D the requirement of this mechanism to induce IFN β . Some of such secretion systems would most likely also be inhibited by cyto D. In vivo direct phagocytosis might not be required. You might want to discuss this.
3. Serum sensitivity: Kocijancic et al. showed serum resistance for ECN but the authors showed convincingly the sensitivity of their strain. The authors might want to point out this discrepancy but it is really only a minor controversy.
4. Figure 5: this might have been a problem of my printer software.
5. Targeting of APCs in vivo: this would require high resolution immune histology to be directly show. However, the reasoning of the authors is acceptable especially since all their functional data point at targeting APCs. The authors might consider this for future work.

Reviewer #2:

Remarks to the Author:

The authors have done a good job responding to the reviewer comments and the manuscript is now much improved. Some minor criticisms remain:

- It is still unclear to me what the benefit is of presenting $n=2$ datasets with mean \pm standard deviation rather than simply just showing the datapoints.
- Fig 2a legend says that these results are representative of at least 3 experiments, but the data presented suggests that $n=2$.
- The datasets in Fig 2h are not consistent in their presentation with ECN-thyA-dapA presented as datapoints + mean (as suggested appropriate above), but the SYN-1891 has Mean \pm SD.
- The error bars for fig 2a and b are not defined in the figure legend.
- Lines 389-396: While it is factually correct that there are more numerous and more efficient tools available for engineering E. coli strains compared to Salmonella, it is not clear that the tools available for Salmonella engineering are sufficiently limited that it would prevent the types of genetic manipulations described in this study. Thus, the statement "attempts to further improve efficacy by engineering in additional functions are often limited due to a lack of tools and methods for genetic manipulation" should be removed or altered accordingly.

We truly appreciate the Reviewers' feedback. Below are our point-by-point responses for the remaining concerns.

Original Reviewer comments appear in black

Our responses appear in green

REVIEWERS' COMMENTS:

Reviewer #1 (Remarks to the Author):

In the revised version of their manuscript Leventhal et al. have appropriately taken care of my concerns. A few minor points need still to be clarified to my mind.

1. In the second paragraph of my original review, I state that their data suggest that the anti-tumor response is induced directly in the neoplasm because they do not find any bacteria outside of the tumor. Presently, it is not known where the anti-tumor reaction is activated. This is important for future work on this type of therapy using bacteria carrying immune modulatory molecules or antigens. Therefore, I requested that this needs to be discussed although I did not state it in the bullet pointed review. Please discuss as I find this is a very important finding.

We appreciate the reviewer's interest in the details of antitumor immune priming and activation following SYN1891 treatment. In particular, the primary question is whether the antitumor T cell activation is initiated in the tumor or at a distant site (for instance within the tumor draining lymph node). While in our study we demonstrate that SYN1891 localization and innate immune activation are restricted to the tumor, we do not formally perform experiments to evaluate where T cell priming occurs. It is known that antigen-presenting cells (APCs) are highly migratory and, depending on their activation state and environmental signals, they circulate between tissue and lymphoid organs. While in one scenario intratumoral antigen presentation and T cell priming might be plausible, the other scenario involves activation of antigen-loaded APCs in the tumor and further translocation of these cells to tumor-draining lymph nodes for T cell activation. Thus, the initiation of immune response by SYN1891 likely happens in the tumor and further development of adaptive immunity could occur in lymphoid organs and/or tumor. In both scenarios, bacteria would not be required to exist outside of the tumor.

2. Phagocytosis: I disagree with the authors although at the moment I cannot prove them wrong. However, my point directs are something different. Bacteria own secretion systems which might target the cytosol of APCs without phagocytosis of the entire bacteria. In vitro they clearly demonstrated bacterial phagocytosis and by using cytochalasin D the requirement of this mechanism to induce IFN β . Some of such secretion systems would most likely also be inhibited by cyto D. In vivo direct phagocytosis might not be required. You might want to discuss this.

The reviewer makes a good and logical point. A wide variety of bacteria do indeed contain various secretion mechanisms, with examples including the production of outer membrane vesicles or the use of injection-like structures such as the Type III secretion system (T3SS). However, unlike pathogenic forms of *E. coli*, our Nissle strain does not endogenously have a T3SS where effectors and potentially some random bacterial dsDNA can be injected into APCs or other cells. It is possible to engineer this system into Nissle. However, the relative complexity of T3SS and impact on manufacturability and stability make it less desirable when considering

the development of a therapeutic. Moreover, throughout our extensive experience working with engineering of *E. coli* Nissle, we have observed a very low level of spontaneous secretion. In fact, in order to achieve meaningful secretion, additional engineering is required. While *E. coli* Nissle does not contain the machinery to actively secrete effectors into APCs, we can not formally rule out the contribution of membrane vesicles which may be recognized by APCs after endocytosis/ fusion *in vivo*. However, the intact bacterial-based therapeutic SYN1891 would likely provide a more biologically relevant context for innate immune activation as it would simultaneously provide bacterial ligands like LPS and double stranded bacterial DNA together with high local concentration of c-di-AMP. In summary, *E. coli* Nissle is not an intracellular, pathogenic bacteria, does not have a secretion system and does not contain any latent virulence factors which would allow it to inject toxins, lysis proteins or other small molecules into the host cells. Therefore, we favor the interpretation that phagocytosis is required for SYN1891's mechanism of action.

3. Serum sensitivity: Kocijancic et al. showed serum resistance for ECN but the authors showed convincingly the sensitivity of their strain. The authors might want to point out this discrepancy, but it is really only a minor controversy.

In our study we formally assess human serum sensitivity of the engineered strain SYN1891, we did not check serum sensitivity of the parental *E. coli* Nissle strain.

4. Figure 5: this might have been a problem of my printer software.

5. Targeting of APCs *in vivo*: this would require high resolution immune histology to be directly show. However, the reasoning of the authors is acceptable especially since all their functional data point at targeting APCs. The authors might consider this for future work.

We thank the reviewer for the comment, we think that development of bacterial and host cell labeling for *in vivo* analysis in combination with flow cytometry techniques will allow us to track both APCs themselves and internalized bacteria in the tumor environment and in draining lymph nodes. We are indeed considering it for future work.

Reviewer #2 (Remarks to the Author):

The authors have done a good job responding to the reviewer comments and the manuscript is now much improved. Some minor criticisms remain.

We want to thank the reviewer for their detailed comments. We have addressed the remaining criticisms in this revision.

- It is still unclear to me what the benefit is of presenting n=2 datasets with mean +/- standard deviation rather than simply just showing the datapoints.

We have removed mean +/- standard deviation and show only datapoints.

- Fig 2a legend says that these results are representative of at least 3 experiments, but the data presented suggests that n=2.

Fig 2a. Data are representative of at least 3 experiments; one experiment is shown with 2 replicates in this experiment per group. This has been clarified within the figure legend.

- The datasets in Fig 2h are not consistent in their presentation with ECN-thyA-dapA presented as datapoints + mean (as suggested appropriate above), but the SYN8-1891 has Mean +/- SD.

As suggested, we have removed mean +/- standard deviation and show only datapoints.

- The error bars for fig 2a and b are not defined in the figure legend.

As suggested, we have removed +/- standard deviation and show only datapoints

- Lines 389-396: While it is factually correct that there are more numerous and more efficient tools available for engineering E. coli strains compared to Salmonella, it is not clear that the tools available for Salmonella engineering are sufficiently limited that it would prevent the types of genetic manipulations described in this study. Thus, the statement “attempts to further improve efficacy by engineering in additional functions are often limited due to a lack of tools and methods for genetic manipulation” should be removed or altered accordingly.

As suggested, we removed our statement.